# A single Na$^+$-P$_i$ cotransporter in *Toxoplasma* plays key roles in phosphate import and control of parasite osmoregulation

**Beejan Asady**[ID][☉], **Claudia F. Dick**[ID][☉], **Karen Ehrenman**[ID][☉], **Tejram Sahu**[ID], **Julia D. Romano**[ID], **Isabelle Coppens**[ID]*

Department of Molecular Microbiology and Immunology, Johns Hopkins University Bloomberg School of Public Health, Baltimore Maryland, United States of America

☉ These authors contributed equally to this work.

* icoppens@jhsph.edu

**Data Availability Statement:** All relevant data are within the manuscript and its Supporting Information files. The raw data of transcriptomic sequencing of WT and ΔPiT strains have been

## Abstract

Inorganic ions such as phosphate, are essential nutrients required for a broad spectrum of cellular functions and regulation. During infection, pathogens must obtain inorganic phosphate (P$_i$) from the host. Despite the essentiality of phosphate for all forms of life, how the intracellular parasite *Toxoplasma gondii* acquires P$_i$ from the host cell is still unknown. In this study, we demonstrated that *Toxoplasma* actively internalizes exogenous P$_i$ by exploiting a gradient of Na$^+$ ions to drive P$_i$ uptake across the plasma membrane. The Na$^+$-dependent phosphate transport mechanism is electrogenic and functionally coupled to a cipargamin sensitive Na$^+$-H$^+$-ATPase. *Toxoplasma* expresses one transmembrane P$_i$ transporter harboring PHO4 binding domains that typify the PiT Family. This transporter named TgPiT, localizes to the plasma membrane, the inward buds of the endosomal organelles termed VAC, and many cytoplasmic vesicles. Upon P$_i$ limitation in the medium, TgPiT is more abundant at the plasma membrane. We genetically ablated the *PiT* gene, and ΔTgPiT parasites are impaired in importing P$_i$ and synthesizing polyphosphates. Interestingly, ΔTgPiT parasites accumulate 4-times more acidocalcisomes, storage organelles for phosphate molecules, as compared to parental parasites. In addition, these mutants have a reduced cell volume, enlarged VAC organelles, defects in calcium storage and a slightly alkaline pH. Overall, these mutants exhibit severe growth defects and have reduced acute virulence in mice. In survival mode, ΔTgPiT parasites upregulate several genes, including those encoding enzymes that cleave or transfer phosphate groups from phosphometabolites, transporters and ions exchangers localized to VAC or acidocalcisomes. Taken together, these findings point to a critical role of TgPiT for P$_i$ supply for *Toxoplasma* and also for protection against osmotic stresses.

## Author summary

Inorganic phosphate (P$_i$) is indispensable for the biosynthesis of key cellular components, and is involved in many metabolic and signaling pathways. Transport across the plasma

deposited in the NIH Gene Expression Omnibus (GEO) database. The accession number is GSE145660. https://www.ncbi.nlm.nih.gov/geo/query/acc.cgi?acc=GSE145660.

**Funding:** CFD recieved grant from the Brazilian National Research Council/CNPq Science without Borders Program (245823/2012-3) 2. http://cienciasemfronteiras.gov.br/web/csf-eng/. IC recieved grant from NIH (AI060767) https://projectreporter.nih.gov/project_info_description.cfm?projectnumber=5R01AI060767-11. The funders had no role in study design, data collection and analysis, decision to publish, or preparation of the manuscript.

**Competing interests:** The authors have declared that no competing interests exist.

membrane is the first step in the utilization of $P_i$. The import mechanism of $P_i$ by the intracellular parasite *Toxoplasma* is unknown. We characterized a transmembrane, high-affinity $Na^+$-$P_i$ cotransporter, named TgPiT, expressed by the parasite at the plasma membrane for $P_i$ uptake. Interestingly, TgPiT is also localized to inward buds of the endosomal VAC organelles and some cytoplasmic vesicles. Loss of TgPiT results in a severe reduction in $P_i$ internalization and polyphosphate levels, but stimulation of the biogenesis of phosphate-enriched acidocalcisomes. ΔTgPiT parasites have a shrunken cell body, enlarged VAC organelles, poor release of stored calcium and a mildly alkaline pH, suggesting a role for TgPiT in the maintenance of overall ionic homeostasis. ΔTgPiT parasites are poorly infectious *in vitro* and in mice. The mutant appears to partially cope with the absence of TgPiT by up-regulating genes coding for ion transporters and enzymes catalyzing phosphate group transfer. Our data highlight a scenario in which the role of TgPiT in $P_i$ and $Na^+$ transport is functionally coupled with osmoregulation activities central to sustain *Toxoplasma* survival.

## Introduction

All organisms depend on an external supply of phosphate to maintain normal growth. Inorganic phosphate ($P_i$) is a key constituent of nucleic acids and membrane phospholipids, and an essential element for energy-mediated metabolic processes and signal transduction pathways in all living organisms [1–7]. The majority of intracellular phosphate ions exist in an organic bound form, as found in phosphate esters, phospholipids, and many phosphorylated intermediate metabolites. For most organisms, the availability of $P_i$ in the environment is a growth limiting factor. The extracellular concentration of $P_i$ is in the micromolar range while its concentration inside cells reaches millimolar values. The higher intracellular concentration of $P_i$ is maintained by $P_i$ transporters located at the plasma membrane [8, 9]. In unicellular eukaryotes, the two main $P_i$ transporter families include the $P_i$ Transporter (PiT) family, which uses either $Na^+$ or $H^+$ to mediate $P_i$ import, and the Phosphate:$H^+$ Symporter (PHS) family [10].

*Toxoplasma gondii* is an obligate intracellular parasite that multiplies in the cytoplasm of mammalian cells within a parasitophorous vacuole (PV). The parasite relies on many host cell metabolites to multiply, and it has evolved efficient strategies to acquire essential nutrients from mammalian host cells. For example, *T. gondii* modifies the permeability of the PV membrane by creating proteinaceous pores that allow small solutes in the host cytosol to enter the PV [11, 12]. Unlike the PV membrane, the parasite's plasma membrane is not freely permeable, and *Toxoplasma* expresses several substrate-specific transporters and translocators at the plasma membrane to internalize nutrients [13–15]. Despite the importance of $P_i$ for the synthesis of numerous phosphorylated metabolic intermediates, nothing is known about the molecular mechanism developed by *Toxoplasma* to import $P_i$ and the dependence of the parasite on external sources of $P_i$ for growth. It has been reported that the parasite accumulates large stores of phosphorus in the form of phosphate ($P_i$), pyrophosphate ($PP_i$) and polyphosphate (polyP), mainly in acidocalcisomes [16]. Selective transporters for $P_i$ and $PP_i$ are likely present on the limiting membrane of acidocalcisomes but have not been identified yet in *Toxoplasma*.

$PP_i$ is a byproduct of many biosynthetic reactions, e.g., synthesis of nucleic acids, coenzymes, proteins, isoprenoids, and activation of fatty acids [17]. $PP_i$ has also bioenergetic roles as they can be generated by photophosphorylation, oxidative phosphorylation and glycolysis,

and used in a number of reactions to replace ATP [18]. Uniquely, *Toxoplasma* possesses higher cellular levels of $PP_i$ than ATP [19]. The concentration of $PP_i$ is regulated predominantly through the activity of soluble pyrophosphatases (PPase) in the cytosol and membrane-bound $H^+$- pumping PPases (V-$H^+$-PPases). *Toxoplasma* expresses a soluble pyrophosphatase (TgPPase), and overexpression of TgPPase in the parasite leads to decreased $PP_i$ concentrations in the cytosol and increased glycolytic flux concomitant to elevated ATP concentrations [20].

In several protozoan parasites, changes in cellular polyP levels are associated with differentiation and stress responses, suggesting an important role of polyP in adaptation to environmental variations [21–23]. A *Toxoplasma* $Ca^{2+}$/$H^+$-ATPase (TgA1) and a vacuolar $H^+$-pyrophosphatase (TgV-$H^+$-PPase) present in acidocalcisomes play roles in regulating intracellular levels of polyP [24, 25]. TgA1-deficient parasites have decreased short- and long-chain polyP content and are less virulent in mice than wild-type *Toxoplasma*. The vacuolar transporter chaperone (VTC) complex produces inorganic polyP by transferring $P_i$ from cytosolic ATP hydrolysis, to acidocalcisomes, and a *T. gondii* homolog (TgVTC2) regulates the stores of long-chain polyP [26]. *Toxoplasma* expresses a phosphate translocator (TgAPT) located to the membranes of the endosymbiotic organelle apicoplast, which internalizes triose phosphate and phosphoenol pyruvate in exchange for $P_i$, and TgAPT is essential for lipoylation of proteins [27].

Acute toxoplasmosis is typically due to reactivation of a latent infection mediated by differentiation of quiescent tissue cysts into rapidly dividing parasites causing focal tissue damage in the brain and muscles. Current treatment options for toxoplasmosis are both limited and poorly tolerated [28]. Recent research has underscored the importance of nutrient salvage pathways for *Toxoplasma* infectivity. Due to the large phylogenic separation between the mammalian host and *T. gondii*, parasite membrane transporters for host metabolite import may constitute valid targets for chemotherapy. In this study, we have characterized the transport mechanism developed by *Toxoplasma* to retrieve $P_i$ from the environment, involving a single $Na^+$-$P_i$ cotransporter, named TgPiT. We successfully deleted the *PiT* gene in *Toxoplasma*, and the phenotypic traits of the resulting mutant reveal a novel role for TgPiT beyond $P_i$ uptake, in participating in osmoregulatory processes, and thus parasite adaptation to its environment.

## Results

### *Toxoplasma* internalizes inorganic phosphate by using a sodium gradient

Inorganic phosphate ($P_i$) is an essential nutrient for all organisms. We investigated the mechanism by which *Toxoplasma* imports $P_i$ from its environment. The kinetics and ion dependence of $P_i$ uptake by *T. gondii* were assayed on extracellular parasites in a phosphate-depleted medium supplemented with radioactive $P_i$, in the presence of either NaCl or $C_5H_{14}ClNO$ (choline chloride) for isotonic replacement of NaCl. *Toxoplasma* incorporates $P_i$ at a rate of $62.2 \pm 10.5$ and $8.1 \pm 1.1$ fmol/min/$10^6$ parasites in the presence of sodium and choline chloride, respectively (Fig 1A). This suggests the requirement of a coupling between $P_i$ and $Na^+$ fluxes across the parasite plasma membrane to drive $P_i$ import. Replacement of $Na^+$ in the medium by $K^+$, $NH_4^+$ or $Li^+$ significantly decreased $P_i$ influx, confirming the involvement of a $Na^+$-dependent transport mechanism for $P_i$ acquisition (Fig 1B). $P_i$ influx shows a sigmoidal dependence on $Na^+$ concentrations (Fig 1C), with a Hill coefficient ($n_H$) of $4.6 \pm 0.4$, $K_{0.5, Na^+}$ values of $3.7 \pm 0.5$ mM and $V_{max}$ values of $572.4 \pm 84$ fmol $P_i$/h/$10^6$ parasites. The sigmoidicity of the curve of $Na^+$ dependence for $P_i$ internalization with a cooperativity index (Hill

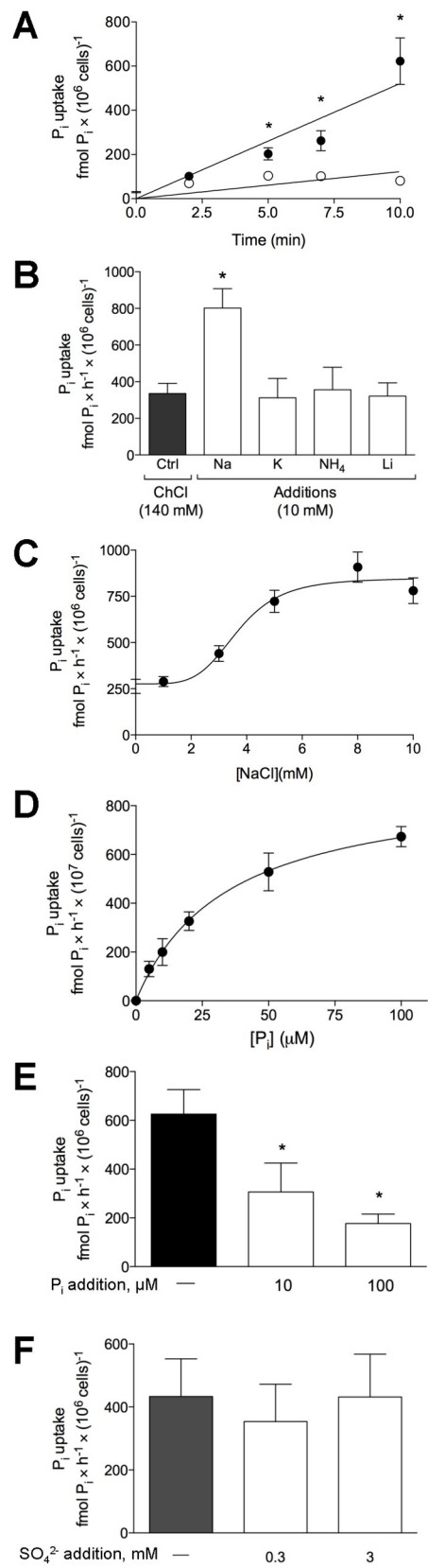

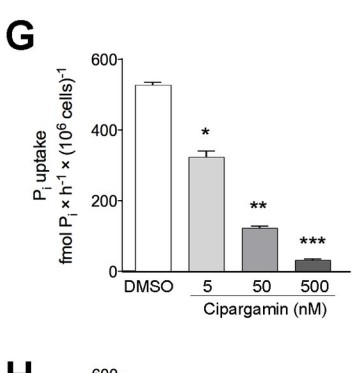

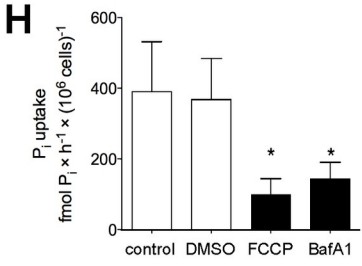

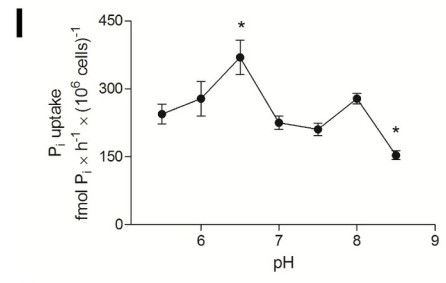

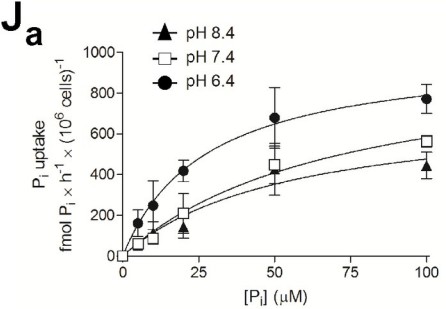

| pH | $V_{max}$ (fmol $P_i$/h/$10^6$parasites) | apparent $K_m$ (µM) | | |
|----|----|----|----|----|
| | | [ $P_i$ ] | [ $H_2PO_4^-$ ] | [ $HPO_4^{-2}$ ] |
| 6.4 | 1006 ± 84 | 31 ± 7 | 12 ± 3 | 17 ± 2 |
| 7.4 | 999 ± 166 | 37 ± 6 | 14 ± 1 | 23 ± 3 |
| 8.4 | 734 ± 92 | 42 ±11 | 11 ± 2 | 31 ± 10 |

**Fig 1. Characteristics of the transport of $P_i$ in *Toxoplasma*. A.** Kinetics of $P_i$ uptake. Extracellular parasites were incubated in a $P_i$-depleted reaction medium at pH 7.4 containing 100 μM $^{32}P_i$ at the indicated times, before washing by filtration and radioactivity counting. Open circles: $Na^+$-independent uptake in the presence of 140 mM choline chloride. Filled circles: $Na^+$-dependent uptake in the presence of 140 mM NaCl. Data are means ± SEM of 4 independent assays. *, $p < 0.05$ (unpaired Student's *t*-test at matched time of reaction). **B.** Ion co-transport activities. $^{32}Pi$ uptake was assayed on parasites incubated in a $P_i$-depleted medium containing radioactive $P_i$ as indicated in A, in the presence of NaCl, KCl, $NH_4Cl$ or LiCl compared to choline chloride (control conditions). Data are means ± SEM (n = 3 independent assays). *, $p < 0.05$ (unpaired Student's *t*-test). **C.** $Na^+$ dependence for $P_i$ uptake. $^{32}P_i$ uptake was assayed on parasites incubated in a $P_i$-depleted medium containing radioactive $P_i$ as indicated in A, with increasing NaCl concentrations. Medium osmolarity was maintained at 300 mOsM with choline chloride supplementation. Data are means ± SEM (n = 6 independent assays). **D.** Saturation curve of $P_i$ transport. $^{32}Pi$ uptake by the parasites was monitored in medium with 140 mM NaCl and various concentrations of $P_i$ at pH 7.4, and traced with 100 μM $^{32}P_i$ (25 μCi/ml assay) for 30 min. Data are means ± SEM (n = 6 independent assays). **E.** Competition assay for $P_i$ transport. $^{32}P_i$ uptake by the parasite was monitored in medium with 140 mM NaCl and excess non-radioactive $P_i$ at pH 7.4, and traced with $^{32}P_i$ as described in D. Data are means ± SEM (n = 3 independent assays). *, $p < 0.05$ (unpaired Student's *t*-test). **F.** Selectivity for $P_i$ influx. Extracellular parasites were incubated in a $P_i$-depleted reaction medium at pH 7.4 containing 100 μM $P_i$, traced with 25 μCi of $^{32}P_i$/ml for 30 min, with added of 0.3, 3 mM $SO_4^{2-}$ or no addition (−). Data are means ± SEM (n = 4 independent assays). **G.** $P_i$ uptake upon inhibition of $Na^+$-$H^+$-ATPase. Extracellular parasites were treated 10 min with 5 nM, 50 nM, 500 nM cipargamin or without drug (DMS0 control) prior to $P_i$ uptake, washed and incubated 10 min in the presence of 100 μM $P_i$ and traced with 250 μCi of $^{32}P_i$ at pH 7.4, in medium containing 140 mM NaCl. Data are means ± SEM (n = 3 independent assays). *, $p = 0.0013$; **, $p = 0.0008$; ***, $p < 0.0001$ comparing with DMSO condition (unpaired Student's *t*-test). **H.** $P_i$ uptake upon condition of described in the presence of 100 μM $P_i$ and traced with 250 μCi of $^{32}P_i$ at pH 7.4, in a medium containing 140 mM NaCl, plus 10 μM FCCP, 100 nM bafilomycin $A_1$, without drug (control) or vehicle (DMSO) for 10 min. Data are means ± SEM (n = 6 independent assays). *, $p < 0.05$ comparing with DMSO (unpaired Student's *t*-test). **I.** $^{32}P_i$ uptake was measured on parasites in a $P_i$-depleted medium containing 140 mM NaCl and traced with 100 μM $^{32}P_i$ (25 μCi/ml assay) for 30 min in different pH ranges. pH in the incubation medium was adjusted by adding concentrated HCl (5.5, 6.0 and 6.5) or KOH (7.0, 7.5, 8.0 and 8.5). The ratio $H_2PO_4^-$/$HPO_4^{2-}$ varied from 50:1 at pH 5.5 to 1:20 at pH 8.5. Data are means ± SEM (n = 4 independent assays). *, $p < 0.05$ (One-way ANOVA using the Tukey's test, comparing each value with that obtained at pH 5.5). **J.** Panel a: pH-dependence of the influx of $^{32}P_i$ into extracellular parasites in medium at pH 6.4, 7.4 or 8.4. Panel b: kinetic constants at pH 6.4, 7.4 or 8.4. The $V_{max}$ and $K_m$ values were extrapolated from the 3 curves in panel a. To determine the $H_2PO_4^-$ and $HPO_4^{2-}$ concentration dependences, the respective concentration in the medium of these ionic forms was calculated at the $P_i$ concentrations shown in panel a, for each pH value: 6.4, 7.4 and 8.4. Asterisks, values obtained at the three different pH values differing significantly from one another ($p < 0.0001$; One-way ANOVA using the Tukey's test).

coefficient) greater than 1 indicates that the transporter has more than one $Na^+$ binding site that function with high cooperativity to drive the internalization of $P_i$.

We next assessed the uptake of $^{32}P_i$ as a function of the concentration of $P_i$ in the medium. The substrate saturation curve of $P_i$ uptake displayed typical Michaelis-Menten behavior with $K_{0.5,\,P_i}$ and $V_{max}$ values of 22.97 ± 5.7 μM and 640.9 ± 14.8 fmol $P_i$/h/$10^7$ parasites, respectively (Fig 1D). Compared to the uptake of $P_i$ by other protozoan parasites [10], these $K_{0.5,\,P_i}$ and $V_{max}$ parameters point to a similar high affinity but lower capacity $P_i$ transport system in *Toxoplasma*. The specificity of the $P_i$ transport mechanism was confirmed by competition with non-radioactive $P_i$ added to the medium as $^{32}P_i$ uptake was reduced proportionally with unlabeled $P_i$ concentrations (Fig 1E). Some $P_i$ transporters exploit inorganic sulfate ($SO_4^{2-}$) as a substrate to promote $P_i$ import into cells [29]. Addition of sulfate to the medium in our uptake assays, however, did not modify $P_i$ influx, suggesting independent transport mechanisms for sulfate and phosphate in *Toxoplasma* (Fig 1F).

To further confirm that *Toxoplasma* utilizes an external gradient of $Na^+$ ions as the driving force for $P_i$ uptake, we examined the effects of the spiroindolone cipargamin, a disruptor of $Na^+$ homeostasis through $Na^+$ efflux inhibition [30]. *Toxoplasma* expresses a $Na^+$-$H^+$-ATPase, named TgATP4 that is localized at the plasma membrane [31]. Pharmacological inhibition of TgATP4 by cipargarmin or reduced TgATP4 expression is detrimental for parasite growth, reflecting the importance of $Na^+$ homeostasis for *Toxoplasma*. The uptake of $^{32}P_i$ by *T. gondii* was monitored following a 10 min-exposure of extracellular parasites to various concentrations of cipargamin. Data show a significant reduction in $P_i$ internalization into parasites, which is proportional to the drug concentration (Fig 1G). As approximately half of the total $P_i$ uptake is driven by an inwardly-oriented $Na^+$ gradient, this result confirms that a primary sustained $Na^+$ gradient is required for the secondary active $Na^+$:$P_i$ transport.

The stoichiometry of $2Na^+:1H_2PO^{4-}$ (or more than two $Na^+$ per one $H_2PO^{4-}$) predicts that the net uptake of $P_i$ is electrogenic, involving an influx of at least one positive charge. To verify this hypothesis, $P_i$ influx was measured under depolarization of the parasite's plasma membrane with either carbonylcyanide-p-trifluoromethoxyphenylhydrazone (FCCP), an iono-phore that collapses membrane potential by activating $H^+$ and $Na^+$ currents, or bafilomycin A1, a selective ATPase inhibitor that target *Toxoplasma* V-$H^+$-ATPase at the plasma membrane [32]. A significant decrease in $P_i$ influx with either FCCP or bafilomycin A1 was observed, which is consistent with the requirement of electrogenic transport for $P_i$ import in *T. gondii* (Fig 1H).

## $P_i$ influx has a higher capacity in an acidic environment

$P_i$ has four ionic species and a pKa of 6.8 under physiological conditions. We next examined whether the uptake of $P_i$ by the parasite varies according to the pH of the medium. The relative proportion of monovalent ($H_2PO_4^-$) is higher than divalent ($HPO_4^{2-}$) $P_i$ anions at more acidic pH values ($< 7.4$) and the opposite at more basic pH values. A rate of $P_i$ influx higher in either an acidic (monovalent) or basic (divalent) pH will inform about the $P_i$ species preference for TgPiT. To determine which ionic form is preferentially imported by *Toxoplasma*, the uptake of $P_i$ at a single concentration (100 µM $P_i$) was monitored in medium with pH ranging from 5.4 to 8.4 (Fig 1I). The rate of $P_i$ uptake at pH 6.4 was approximately two-fold higher than at pH 8.4. Then, we investigated the $P_i$ uptake as a function of a range of $P_i$ concentrations at the pH values of 6.4, 7.4 and 8.4. Results confirm that $P_i$ internalization rates were higher at pH 6.4 than at the other pH values (Fig 1H, panel a). These data were used to calculate the kinetic parameters $V_{max}$ and $K_m$ (Fig 1J, panel b). Plotting $P_i$ influx as a function of the $H_2PO_4^-$ concentrations results in similar apparent $K_m$ values of $P_i$ uptake at pH 6.4, 7.4 and 8.4. When the same analysis was applied for $HPO_4^{2-}$, the apparent $K_m$ values diverged between the three pH values, with the lowest value obtained at pH 6.4. These data suggest that *Toxoplasma* may have a strong preference for $H_2PO_4^-$ over $HPO_4^{2-}$ at physiological pH, as the apparent variation of $K_m$ ($P_i$) with pH reflects the abundance of $H_2PO_4^-$.

## *Toxoplasma* expresses a $P_i$ transporter homologue of the PiT family

The annotated *T. gondii* genome has a single putative phosphate transporter family protein (ToxoDB accession #: TGGT1_240210). Based on sequence comparison and domain identification, the parasite transporter bears closer similarity to the $P_i$ transporters of fungi and animals: it harbors PHO4 binding domains identified in the PHO4 protein of the filamentous fungus *Neurospora crassa* [33] that typifies the PHO89 proteins of the PiT family (Fig 2A and 2B). PHO89 proteins consist of 10–12 transmembrane domains (TMD), with both N- and C-termini exposed to extracellularly, and a large intracellular hydrophilic loop positioned between TMD 6 and 7 (in 10 TMD) or TMD 7 and 8 (in 12 TMD) [34–36]. The *T. gondii* $P_i$ transporter, designated here as TgPiT, has 869 aa with a predicted size of 92-kDa, and shares the topological features of PHO89 proteins (Fig 2B). TgPiT shows 38%, 40% and 49% identity to the human, *Saccharomyces cerevisiae* and *Plasmodium falciparum* homologues, respectively [6, 37]. Functional studies on PHO89 from *S. cerevisiae* demonstrate that two glutamic acid residues at positions 55 and 490, conserved in all PiT family members and located in the PHO4 domains, are essential for $P_i$ transport activity in yeast [34, 38]. Based on sequence alignments, the equivalent glutamic acid residues in TgPiT are located at positions 68 and 763 on the parasite PHO4 domains at the N- and C-termini (S1 Fig).

To examine the expression of TgPiT in *T. gondii*, we generated an antibody against a recombinant peptide from aa$_{275-417}$ of the TgPiT sequence. Western blots of parasite lysates

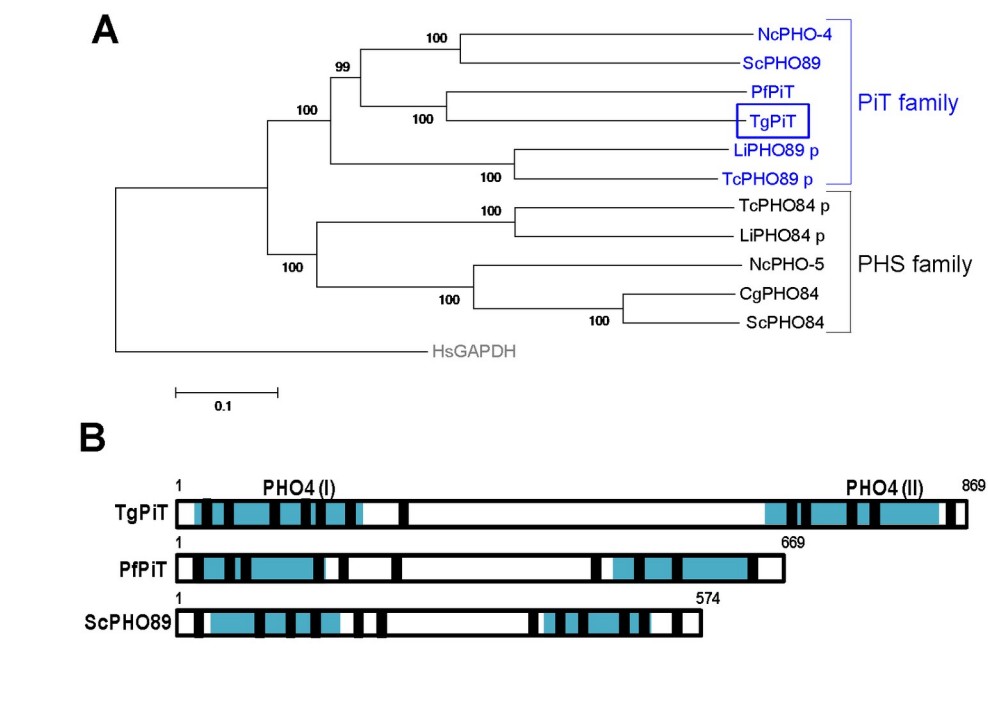

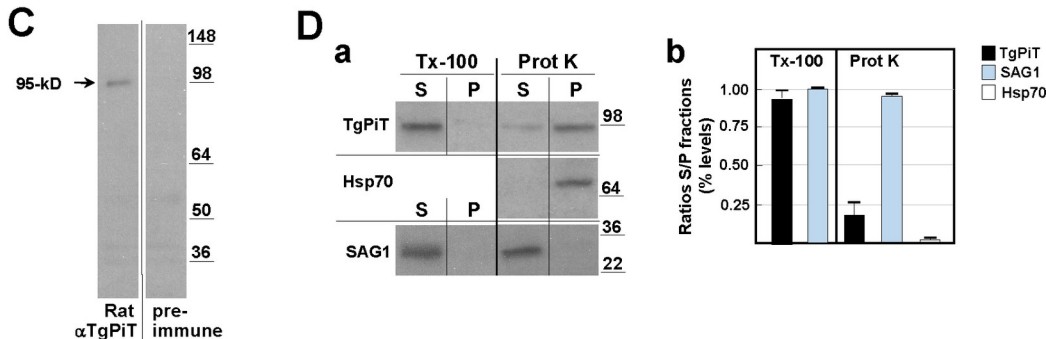

**Fig 2. Motif features and expression of TgPiT in *Toxoplasma*. A.** Phylogenetic analysis of the PHS and PiT family proteins. Amino acid sequences from different unicellular eukaryotic species were aligned and phylogenetic analysis was performed using MEGA 5.2.2 software. The PiT family members include PHO-4 from *Neurospora crassa* (GenBank: AAA33607.1), ScPHO89 from *Saccharomyces cerevisiae* (GenBank: NP_009855.1), PfPiT from *Plasmodium falciparum* (GenBank: CAE30463.1), LiPHO89 p from *Leishmania infantum* (GenBank: XP_001466587.1) and TcPHO89 p from *Trypanosoma cruzi* (GenBank: XP_813912.1), The PHS family members include TcPHO84 p from *T. cruzi* (GenBank: XM_809326.1), LiPHO84 p from *L. infantum* (GenBank:AFJ96967.1), NcPHO-5 from *N. crassa* (GenBank: AAA74899.1), CgPHO84 from *Candida glabrata* (GenBank: XM_445078.1) and ScPHO84 from *S. cerevisiae* (GenBank: CAA89157.1). p = putative sequence based on functional motif identifications. Outgroup: HsGAPDH from *Homo sapiens* (GenBank: NP_002037.2). **B.** Conserved domains in TgPiT. TgPiT has two conserved PHO4 domains (I and II) containing GLU residues required for $P_i$ translocation (turquoise boxes), present in the PiT of yeast (ScPHO89) and *P. falciparum* (PfPiT; MAL13P1.206). TgPiT contains 12 putative TMD (black boxes) and a large intracellular loop at between TMD 7 and 8. **C.** Expression of TgPiT. Immunoblots of *Toxoplasma* lysates (10E7 parasites per lane) were incubated with anti-TgPiT antibodies showing a band at ~95 kD. **D.** Biochemical analysis of TgPiT in isolated parasites. Solubilization of TgPiT: after washing, parasites isolated from cells were lysed in buffer containing 1% TritonX-100 (Tx-100) for 15 min before centrifugation of the lysate and collection of the supernatant (S; detergent-solubilized fraction) and pellet (P; membrane fraction) for SDS-PAGE and Western blotting using antibodies against TgPiT or the surface protein SAG1 as positive control (panel a). Surface-exposure of TgPiT: after washing, extracellular *Toxoplasma* were incubated 30 min in the presence of 0.1 mg/ml of proteinase K (Prot K) or reaction buffer alone at 22°C before adding PMSF to inactivate proteinase K, centrifugation to collect the supernatant (S; Proteinase K-sensitive fraction) and the pellet (P; Proteinase K-resistant fraction) for SDS-PAGE and Western blotting using antibodies against TgPiT, SAG1 (plasma membrane) and Hsp70 (cytosol) antibodies as controls for surface-proteolysis (panel a). Panel b shows the quantification of the ECL signal on immunoblots from 3 independent assays (means ± SD) and expressed in percent of ratios of S fractions to P fractions for TritonX-100 or proteinase K assays.

show a band corresponding to ~95-kD (Fig 2C). The solubility properties of TgPiT were then examined after protein extraction in the presence of the non-denaturing detergent TritonX-100, and TgPiT was mainly recovered in the detergent-solubilized fraction, in accordance to the presence of potential TMD of the parasite transporter (Fig 2D, panels a and b). To examine whether TgPiT is located at the plasma membrane of *T. gondii*, we treated parasites with proteinase K that nonselectively digests surface-exposed proteins. After centrifugation to separate the proteinase K-sensitive fraction in the supernatant (parasite surface) from the proteinase K-resistant fraction in the pellet (parasite interior), ~80% of TgPiT material was detected in the pellet. This indicates that the transporter is mainly present within the parasite, on organellar membranes (Fig 2D, panels a and b), which contrasts to other PiT family members that are expressed at the plasma membrane.

## TgPiT localizes to the VAC compartment, cytoplasmic vesicles and the plasma membrane

We conducted fluorescence microscopy studies to scrutinize the localization of TgPiT in *Toxoplasma*. In a first approach, *Toxoplasma* were transfected with a plasmid containing TgPiT fused to mCherry, and the fluorescence signal on intracellular parasites was detected on several, well-defined structures throughout the cytoplasm (Fig 3A, panel a). In a second assay, extracellular *Toxoplasma* were immunostained with antibodies against TgPiT for immunofluorescence assays (IFA), and punctate fluorescent foci were also observed inside the parasite (Fig 3A, panel b). In addition, anti-TgPiT antibodies detected a large structure at the apical end of extracellular parasites as well as large sparse area on the plasma membrane. The intracytoplasmic punctate pattern for TgPiT could be reminiscent of secretory dense granule organelles that are distributed throughout the cytoplasm; however, a double staining of *Toxoplasma* using anti-GRA7 (as a marker of dense granules) and anti-TgPiT antibodies showed no colocalization of GRA7 and TgPiT signals regardless of PV size (S2 Fig). The spotty fluorescence signal at the parasite periphery could correspond to patches on the plasma membrane. We performed dual staining IFA on extracellular parasites using antibodies against TgPiT and SAG1 (a marker of the plasma membrane), and in some instances, the TgPiT and SAG1 signals colocalize, indicating TgPiT distribution at the plasma membrane. In addition, several TgPiT-containing structures were observed close to the SAG1 signal, aligned beneath the plasma membrane (Fig 3B).

To determine the nature of organelles containing TgPiT, we performed immunoEM using anti-TgPiT antibodies on intracellular *Toxoplasma* 24 h post-invasion (p.i.). Gold particles were observed on the apical endolysosomal compartment, termed Vacuolar Compartment (VAC) or Plant-Like Vacuole (PLV) (hereafter referred to as VAC) [39, 40] that undergoes fragmentation into small vesicles throughout the cytoplasm during replication (Fig 3C). The VAC is characterized by the presence of internal vesicles formed by the inward budding of the limiting membrane, as found in mammalian late endosomal multivesicular bodies (MVB). Interestingly, TgPiT was selectively distributed on the membranes of intraluminal vesicles of the VAC. In addition, TgPiT was detected on several cytoplasmic vesicles (Fig 3D), some close to the plasma membrane (Fig 3E), and in patches on the plasma membrane (Fig 3D–3F).

## Upon phosphate limitation in the medium, *Toxoplasma* exports TgPiT to the plasma membrane

Phosphate concentrations in the serum and within mammalian cells vary with diet [41], and such a fluctuation may affect the intracellular development of *Toxoplasma*. In some cells, phosphate supplementation in the medium results in a decrease in the plasma membrane

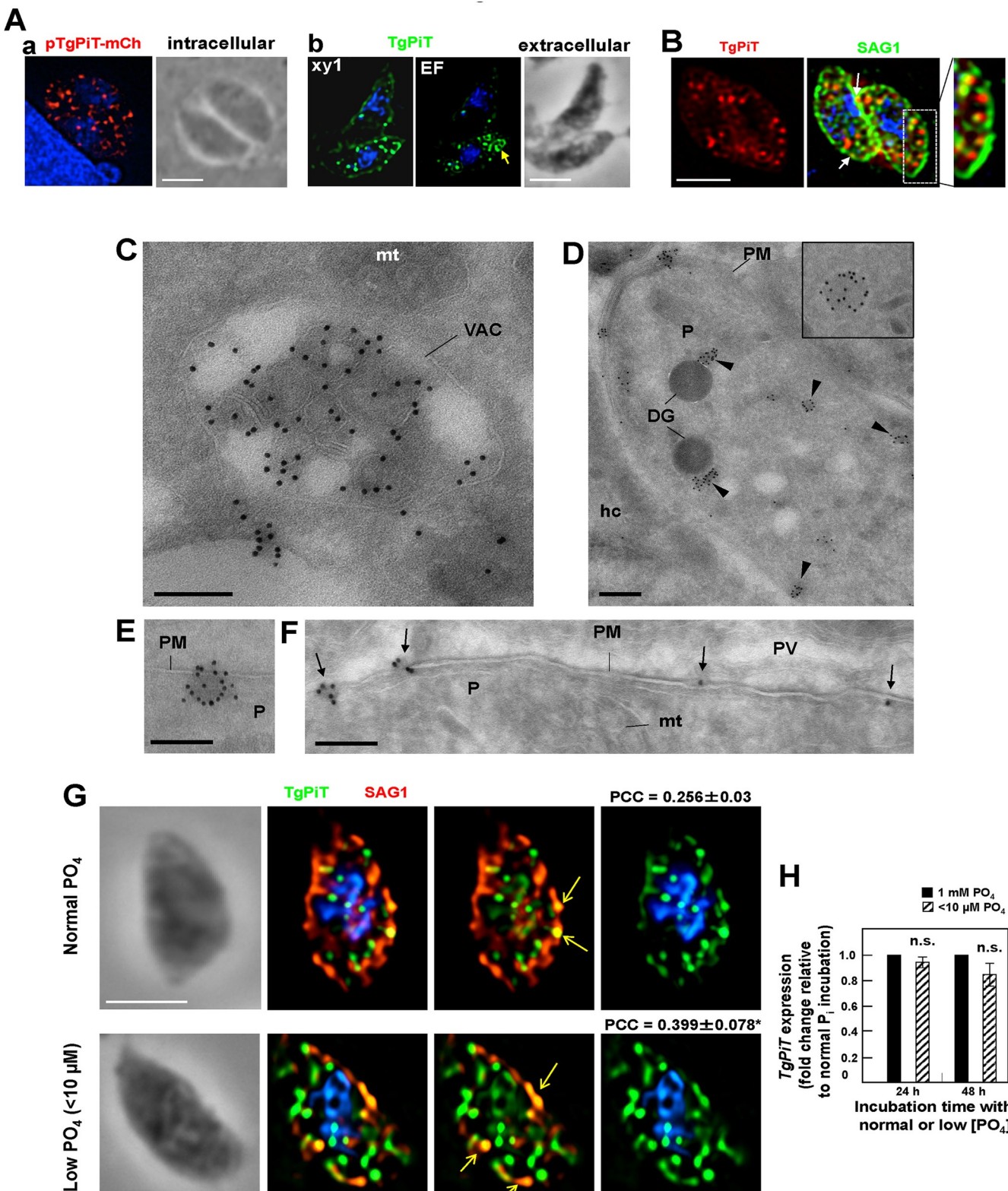

**Fig 3. Localization of TgPiT in *Toxoplasma*. A.** Panel a: Fluorescence microscopy of intracellular *Toxoplasma* transfected with a plasmid containing TgPiT-mCherry and visualized 16 h post-transfection. An individual *z*-slice is shown. Panel b: Fluorescence microscopy of extracellular *Toxoplasma* immunostained with rat anti-TgPiT antibodies. EF, extended focus, xy1, single *z*-slice. Both panels show intraparasitic puncta for TgPiT. Arrow in panel b points to a larger

structure. Scale bars, 5 μm. **B.** Double IFA using anti-TgPiT and anti-SAG1 antibodies, showing TgPiT signal aligned along the plasma membrane (inset) or at the plasma membrane (arrows). An individual z-slice is shown. Scale bar, 5 μm. **C-F.** ImmunoEM of *Toxoplasma*-infected fibroblasts for 24 h using anti-TgPiT antibodies revealed by IgG-gold particles showing TgPiT on the VAC compartments (in C), various vesicles distributed throughout the cytoplasm (in D, arrowheads) or close to the plasma membrane (PM, in E) and at the plasma membrane (C, D, F). DG, dense granules; hc, host cell; mt, mitochondrion; P, parasite. Scale bars, 300 nm. **G.** IFA on extracellular *Toxoplasma* using anti-TgPiT and anti-SAG1 antibodies. Prior to the IFA, intracellular parasites have been incubated under normal culture conditions with 1 mM phosphate ($PO_4$) or in phosphate-poor medium with <10 μM $PO_4$ for 24 h before isolation. Arrows show patches of TgPiT co-localizing with SAG1 that are more pronounced under condition of low $PO_4$ conditions. Representative images are show, from 50–65 parasites. Individual z-slices are shown. The Pearson's correlation coefficient (PCC) was calculated based on the fluorescent signal on the whole parasite. PCC values are means ± SD; *, $p<0.05$ (unpaired Student's *t*-test). **H.** Real-Time qPCR for transcriptional analyses for TgPiT at 1 mM $PO_4$ or <10 μM $PO_4$ at the indicated times. Data of TgPiT was normalized to parasite α-actin housekeeping gene to calculate $2^{-\Delta\Delta CT}$ values. Results are graphed as folds of induction, normalized to the TgPiT transcripts at 1 mM $PO_4$ condition. Means ± SD, n = 4 independent assays in triplicate, no significant difference (unpaired Student's *t*-test). Scale bars, 5 μm.

expression of $P_i$ transporters while phosphate deprivation leads to their up-regulation [42, 43]. If TgPiT is the main transporter for $P_i$ internalization into *Toxoplasma*, the parasite may adapt to varying extracellular phosphate concentrations by modulating the expression and localization of TgPiT. Under normal culture conditions, the phosphate concentration is ~1 mM in the culture medium. To mimic a phosphate deficiency condition, we incubated intracellular parasites in a phosphate-free medium supplemented with 10% FBS, resulting in a total phosphate concentration less than 10 μM. After 2 days of cultivation of *Toxoplasma*-infected fibroblasts in the presence of normal (1 mM) or low (< 10 μM) phosphate, parasites were isolated from cells and fixed for IFA for TgPiT and SAG1 localization. Under the low phosphate condition, the signal for TgPiT became more pronounced at the plasma membrane, as evidenced by relatively higher Pearson correlation coefficient (PCC) values between the TgPiT and SAG1 signals (Fig 3G).

To determine if this adjustment in TgPiT localization in response to low phosphate correlated with changes in TgPiT expression, we performed quantitative RT-PCR on intracellular parasites maintained under conditions of phosphate starvation for 1 and 2 days. In comparison with normal culture conditions, no significant changes were observed in TgPiT transcript levels in phosphate-starved parasites (Fig 3H). This suggests that *Toxoplasma* is solely able to respond to phosphate deprivation in the medium by demobilizing TgPiT from internal stores to the plasma membrane, not by up-regulating the expression of TgPiT.

## ΔTgPiT parasites exhibit replication delay in cultured cells

We next generated a *Toxoplasma* strain lacking the *PiT* gene to assess the functional importance of TgPiT in sensing and/or supplying phosphate to the parasite. To genetically ablate the *PiT* gene via recombination and insertion of the HXGPRT selectable marker cassette, CRISPR/Cas9 technology was used, and viable clones were obtained (S3A Fig). The deletion of *PiT* and insertion of the HXGPRT cassette were verified at the genomic level using specific primers (S3B Fig), and at the protein level using anti-TgPiT antibodies on immunoblots of parasite lysates (S3C Fig) and on PFA-fixed parasites (S3D Fig), and no signal was detected with anti-TgPiT antibodies in knockout parasites.

To investigate the phenotype of ΔTgPiT parasites, we first measured replication rate at 24 h p.i. by enumeration of parasites per PV. While most parental PV contained 4 to 8 parasites, the majority of ΔTgPiT PV had only 2 to 4 parasites (Fig 4A). This replication delay was confirmed using radioactive uracil incorporation assays, showing ~30% less uracil associated with ΔTgPiT parasites (Fig 4B). This replication defect was accompanied by morphology abnormalities in the mutant parasites. IFA using anti-SAG1 antibodies on intracellular *Toxoplasma* were performed to inspect the global shape of mutant parasites. While 16% of PV for ΔTgPiT parasites had similar morphology to the parental strain with healthy-looking parasites organized in a rosette within the PV, 23% of PV from knockout parasites harbored a mixed

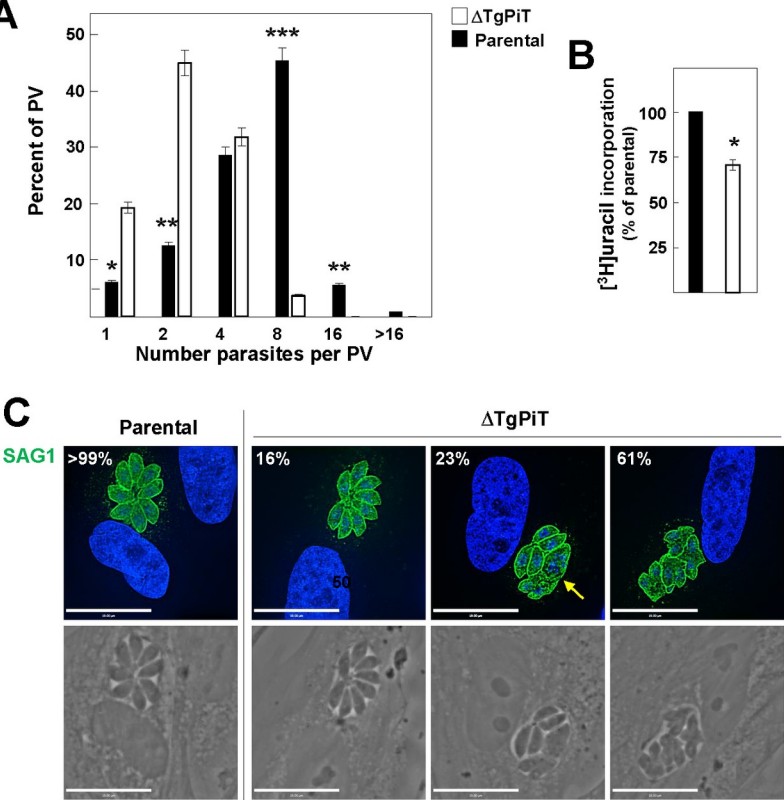

**Fig 4. Replication and growth of ΔTgPiT parasites. A-B.** Quantitative measurement of the replication rate of parental and ΔTgPiT parasites 24 h p.i. assessed by parasite counting per PV (A) or [³H]uracil incorporation assays (B), showing replication delay for the knockout. Data in A and B are means ± SD, n = 3 independent assays. *, $p < 0.05$; **, $p < 0.01$; ***, $p < 0.005$; (unpaired Student's $t$-test). **C.** IFA of infected HFF for 24 h with parental or ΔTgPiT parasites with anti-SAG1 antibodies showing % representative images of 40–60 PV, revealing aberrant cell shape of the knockout (arrow). Individual *z*-slices are shown.

population of parasites with either normal or aberrant morphology, and 61% of the PV showed only distorted parasites (Fig 4C).

To ascertain that the developmental defect of ΔTgPiT parasites was caused by the loss of *PiT*, we reintroduced the *PiT* gene into ΔTgPiT parasites and confirmed this re-insertion at the genomic level with specific primers (S4A and S4B Fig). The expression of the *PiT* gene in the complemented strain (ΔTgPiT::*PiT*) was driven by the uracil phosphoribosyl transferase (UPRT) promotor, and complemented parasites exhibited a 2-fold higher expression level of TgPiT transcripts than parental (S4C Fig). Western blotting and IFA using anti-TgPiT antibodies on ΔTgPiT::*PiT* parasites confirm the expression of TgPiT and its correct localization (S5A and S5B Fig). The replication phenotype assessed at 24 h p.i. was also restored to normal levels in ΔTgPiT::*PiT* parasites, and both complemented and parental parasites began to egress from host cells 48 h p.i. as expected (S5C Fig). Replication rates assessed by counting parasites per PV were similar between the two strains (S5D Fig). Growth monitored by plaque assays reveals no difference in plaque number and size between parental and ΔTgPiT::*PiT* parasites (S5E Fig).

## ΔTgPiT parasites show reduced acute virulence in mice

We determined the acute virulence of ΔTgPiT parasites in a murine model. Outbred mice were infected with an intravenous or subcutaneous inoculum of 150 and 50 parasites,

respectively, from the parental, ΔTgPiT or ΔTgPiT::*PiT* strains, and PBS as control for mice. Mice receiving parental or complemented parasites by intravenous inoculation showed mortality starting at day 8 post-inoculation, and all mice had expired by day 9 (Fig 5A, panel a). By contrast, 80% of mice infected with the ΔTgPiT parasites were still alive at day 15 but they all succumbed the next day. Despite survival to 14–15 days, mouse cachexia was evident as revealed by the gradual weight loss of ΔTgPiT-infected mice until death, a sign of feebleness as the infection progressed (Fig 5A, panel b). Plaque assays of fibroblasts infected with the same batches and number of parental, ΔTgPiT or ΔTgPiT::*PiT* parasites used to infect mice were in accordance with the percent values of mouse survival (Fig 5A, panel c).

All mice infected subcutaneously with parental or complemented parasites died by day 12. In contrast, one mouse infected with ΔTgPiT parasites died on day 21, and 33% of mice were still alive at day 36 although displaying signs of cachexia based on weight loss (Fig 5B, panels a and b). Plaque assays of fibroblasts infected with 50 parasites for 9 days corresponded with the *in vivo* acute infection results (Fig 5B, panel c). Given the hypervirulence of the Type I *Toxoplasma* strain in murine models, the ΔTgPiT mutant dramatically lessened its acute virulence, indicating that the TgPiT protein is required for optimal virulence of *Toxoplasma* in vivo.

## ΔTgPiT parasites incorporate lower amounts of $P_i$

We next examined the contribution of TgPiT to importing $P_i$ in *Toxoplasma* by assessing the $P_i$ uptake ability of ΔTgPiT parasites. Extracellular ΔTgPiT parasites were exposed to $^{32}P_i$ for 2 or 10 min to monitor the radioactive $P_i$ incorporated into the parasite, in the presence of sodium or in a sodium-free solution (choline chloride). In comparison to parental and ΔTgPiT::*PiT* parasites, a drastic reduction in the amount of exogenous $P_i$ associated with the mutant was observed at both times, with no significant difference between the conditions with or without sodium in the transport assay medium (Table 1). Similar to parental parasites, PiT-complemented parasites showed significantly higher uptake of $P_i$ in the presence of NaCl. These data indicate that *Toxoplasma* relies mainly on TgPiT as a supplier of $P_i$.

## ΔTgPiT parasites contain lower levels of polyP and free $P_i$

The poor ability of ΔTgPiT parasites to acquire essential $P_i$ from the phosphate-rich environment may result in decreased intraparasitic levels of phosphate, and thus deficits in the synthesis of many phosphorylated metabolites. The content of $P_i$, free or in the form of short and long polyP polymers, was measured in ΔTgPiT parasites, and compared to that in parental and ΔTgPiT::*PiT* parasites. In $HClO_4$ extracts of parasites incubated with exopolyphosphatase, the total $P_i$ concentration was reduced by 2-fold in the mutant compared to the parental strain, with the concentration of polyP decreased by ~45% (Fig 6). Interestingly, the concentrations of polyP and free $P_i$ were 1.5- and 1.7-times higher, respectively, in ΔTgPiT::*PiT* parasites compared to parental parasites. The greater amount of phosphate in the complemented strain may be attributable to the 2-fold increase in *PiT* transcripts levels. These data suggest that the impaired ability of ΔTgPiT parasites to import $P_i$ results in a drastic reduction in polyP stores that may be deleterious for the parasite intracellular development.

## ΔTgPiT parasites have a reduced cell volume

We next examined the ultrastructure of intracellular ΔTgPiT parasites by transmission EM, in comparison to parental parasites 24 h p.i. We confirmed that the majority of PV contained very few ΔTgPiT parasites while parental parasites had replicated ~3 times, forming large rosettes (Fig 7A, panels a and b). A first noticeable difference was the odd shape and slenderness of the cell body of ΔTgPiT parasites. We next quantified the cell volume of ΔTgPiT

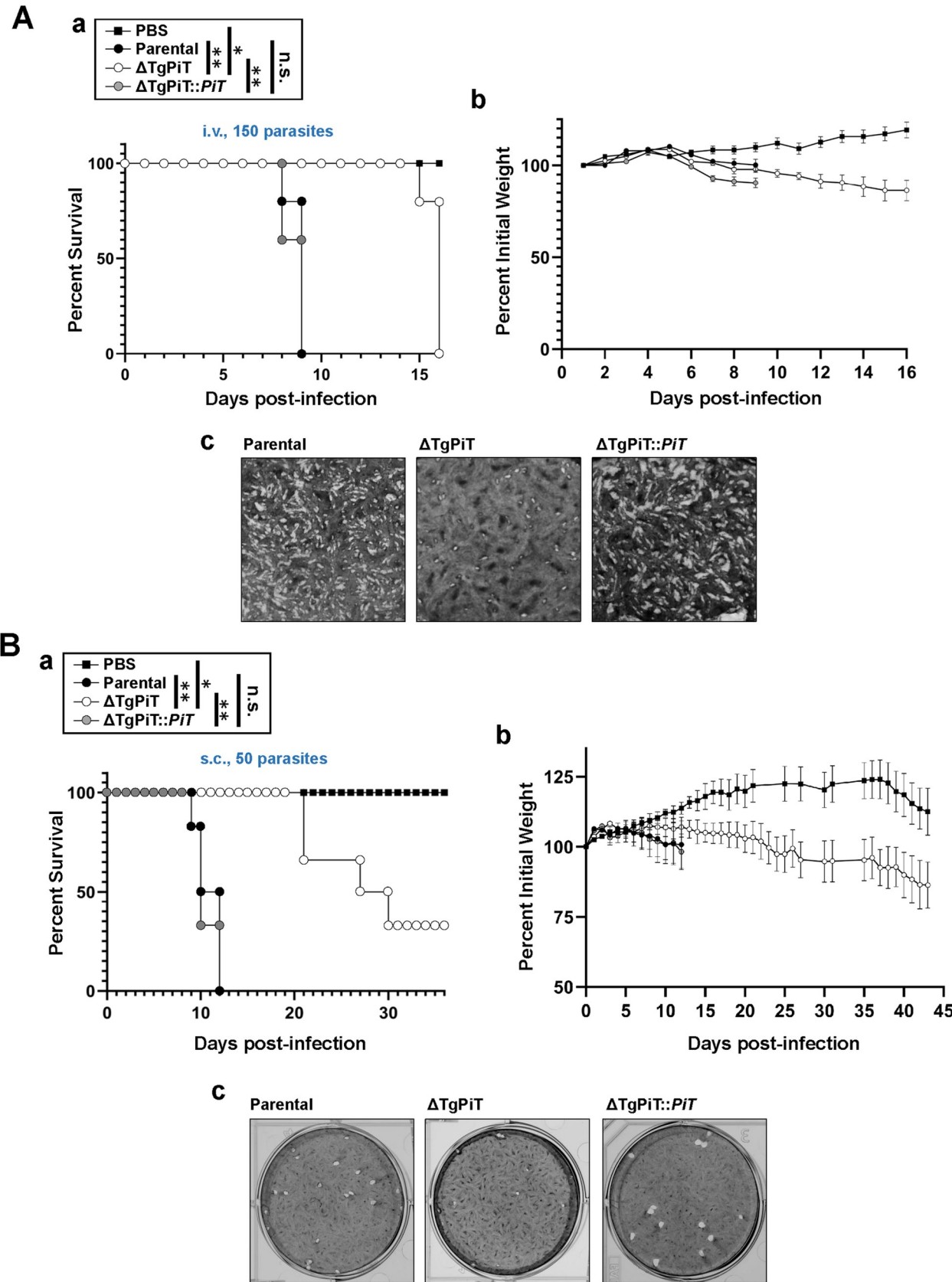

**Fig 5. Growth features of ΔTgPiT in vivo.** The acute virulence of ΔTgPiT parasites was evaluated in a murine model via two routes of infections. **A.** 150 parasites from each strain or PBS alone were used to infect intravenously outbred mice (n = 5 mice for each strain), and the mortality (panel a) and weight (panel b) of the mice were monitored daily. Panel c shows a magnified view of each plaque assay in which monolayers of fibroblasts were infected with 150 parasites for 5 days. *, $p = 0.0024$; **, $p < 0.0001$ (Log-rank Mantel-Cox test). **B.** 50 parasites from each strain or PBS alone were used to infect subcutaneously outbred mice (n = 6 mice for each strain) to monitor the mortality (panel a) and weight (panel b) of the mice. Note that Y axis starts at 50% for better representation of the data for each group. Plaque assays on infected fibroblasts with 50 parasites for 9 days are shown in panel c. *, $p = 0.0496$; **, $p < 0.0001$ (Log-rank Mantel-Cox test).

parasites compared to parental and complemented parasites, using a Coulter counter and a gating strategy in which parasite numbers were assessed based on their cell volume [44]. Parasites were gated from volumes ranging from 10 to 40 $\mu m^3$. One hundred percent of parasites from all 3 strains had a cell volume greater than 10 $\mu m^3$. Very few ($<5\%$) ΔTgPiT parasites with a volume greater than 25 $\mu m^3$ were detected while over 90% of parental and complemented parasites had a volume $\geq 25$ $\mu m^3$ (Fig 7B, panel a). For quantification, curves for every biological replicate (n = 3) of each strain were generated, highlighting a decline in parasite number as the gating volume increases (S6 Fig). From these curves, a linear regression analysis was applied and the parameters of the resulting equations were used to calculate the average size of parasites from each strain (Fig 7B, panel b). Data indicate an average cell volume of 19 $\mu m^3$ for knockout parasites and 32 $\mu m^3$ for both parental and complemented parasites (Fig 7B, panel b).

## Acidocalcisome biogenesis is stimulated in ΔTgPiT parasites

A second observation was the presence of abundant acidocalcisomal profiles in the cytoplasm of ΔTgPiT parasites (Fig 8A, arrowheads in panel a vs. panel b). Acidocalcisomes are acidic calcium and phosphate storage organelles conserved from bacteria to humans, which have an acidic matrix containing several cations bound to phosphates, mainly present in the form of

**Table 1. $P_i$ uptake by parental, ΔTgPiT and ΔTgPiT::*PiT* parasites** Intracellular parasites were cultivated for 24 h in $P_i$-depleted DMEM with 1% FBS prior to the phosphate uptake assay, which was performed as described in legend of Fig 1, in a $P_i$-depleted reaction medium at pH 7.4 containing 100 $\mu M$ $^{32}P_i$ in the presence of 140 mM NaCl or choline chloride (ChCl) for energized and unenergized phosphate uptake conditions, respectively, at the indicated times.

| Conditions | $P_i$ uptake (fmol $P_i$ x ($10^6$ cells)$^{-1}$ (mean ± SEM, n = 3) |
|---|:---:|
| **2 min, NaCl** | |
| a. Parental | 79.2 ± 40.5 |
| b. ΔTgPiT | 6.3 ± 2.7 |
| c. ΔTgPiT::*PiT* | 79.9 ± 48.3 |
| **2 min, ChCl** | |
| a'. Parental | 24.9 ± 14.2 |
| b'. ΔTgPiT | 4.0 ± 1.6 |
| c'. ΔTgPiT::*PiT* | 6.2 ± 2.4 |
| **10 min, NaCl** | |
| d. Parental | 525.6 ± 123.6 |
| e. ΔTgPiT | 86.9 ± 36.9 |
| f. ΔTgPiT::*PiT* | 274.0 ± 57.9 |
| **10 min, ChCl** | |
| d'. Parental | 118.7 ± 58.6 |
| e'. ΔTgPiT | 62.0 ± 27.6 |
| f'. ΔTgPiT::*PiT* | 171.2 ± 21.9 |

The $p$ values are $p < 0.015$ for conditions **f** versus **f'**; $p < 0.004$ for conditions **a** versus **a'** and **c** versus **c'**; $p < 0.0001$ for conditions **a** versus **b**, **d** versus **e** and **d** versus **d'**. The $p$ values are not significant for conditions **b** versus **b'** and **e** versus **e'** (unpaired Student's *t*-test).

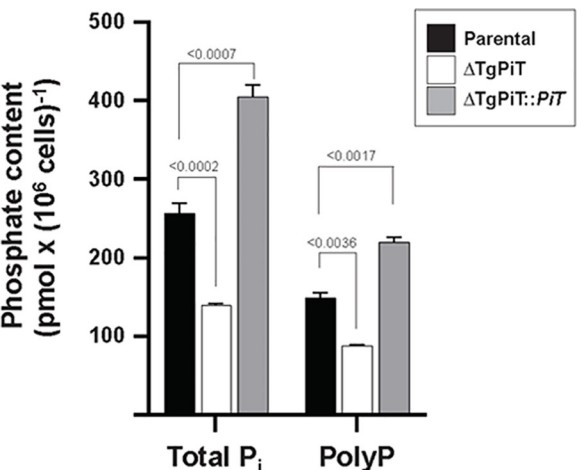

**Fig 6. Determination of phosphate content in ΔTgPiT.** Extracellular parental, ΔTgPiT and ΔTgPiT::*PiT* parasites treated with perchloric acid were incubated with active or inactive exopolyphosphatase to measure inorganic phosphate. Total $P_i$ corresponds to free monomeric $P_i$ and polyP. The concentrations of polyP were deduced from values obtained on exopolyphosphatase-treated samples (giving total $P_i$ concentrations) subtracted from values obtained on denatured exopolyphosphatase-treated samples (giving free Pi concentrations). Values are mean ± SD, n = 3 independent assays. *p* values were calculated using Fisher's LSD test.

short and long chain polyP [16]. Acidocalcisomes play an important role in pH homeostasis and osmoregulation. By EM, acidocalcisomes are characterized by their roundish shape, a thin layer of electron-dense material that sticks to the inner face of the membrane, and e-lucent matrix containing high electron-dense inclusions, whose size varies depending on deposits of phosphorus and calcium. Acidocalcisomes in the ΔTgPiT parasites were not only plentifully present both at the apical end (Fig 8B, panel a) and basal end (Fig 8B, panel b), but were also particularly large. Their luminal inclusions were variable in morphology, and could occupy their entire matrix (Fig 8C, panels i to vii). The shapes of acidocalcisomes in the mutant also varied, from spherical (Fig 8C), elongated (Fig 8D, panel i) to lobulated (Fig 8D, panels ii to iv). These observations collectively suggest that perturbations in phosphate mobilization could lead to alterations of the acidocalcisome morphology with abnormal deposits of phosphorous and/or calcium. Finally, EM sections illustrate close connection between acidocalcisomes and other organelles such as the mitochondrion (Fig 8E, panel a), ER tubules (Fig 8E, panel b), or the VAC (Fig 8E, panel c), as previously described [45].

Acidocalcisomes can be directly observed on whole mounts of cells (unfixed, unstained and air-dried), deposited onto formvar-coated grids, in which they appear by EM as clearly delineated electron-dense spherical structures within the cytoplasm. We quantified the acidocalcisomal population in ΔTgPiT parasites in comparison to parental and complemented parasites by EM observations of entire parasites applied to grids. We confirmed the massive accumulation of acidocalcisomes in ΔTgPiT parasites, at various sizes and sometimes in clusters (Fig 9A). Enumeration of acidocalcisomes per parasite reveals an average of 7 and 9 for parental and ΔTgPiT::*PiT* parasites, respectively, while the mutant contained up to ~30 on average (Fig 9B). Maintenance of parasites in low phosphate medium for 24 h significantly reduced by 2-fold the number of acidocalcisomes in ΔTgPiT parasites. These data suggest that ΔTgPiT parasites may compensate for the reduction of exogenous $P_i$ internalization by sequestering phosphate, likely liberated from internal molecules, in acidocalcisomes. Under condition of reduced $P_i$ availability in the environment, this compensatory pathway may be down-regulated

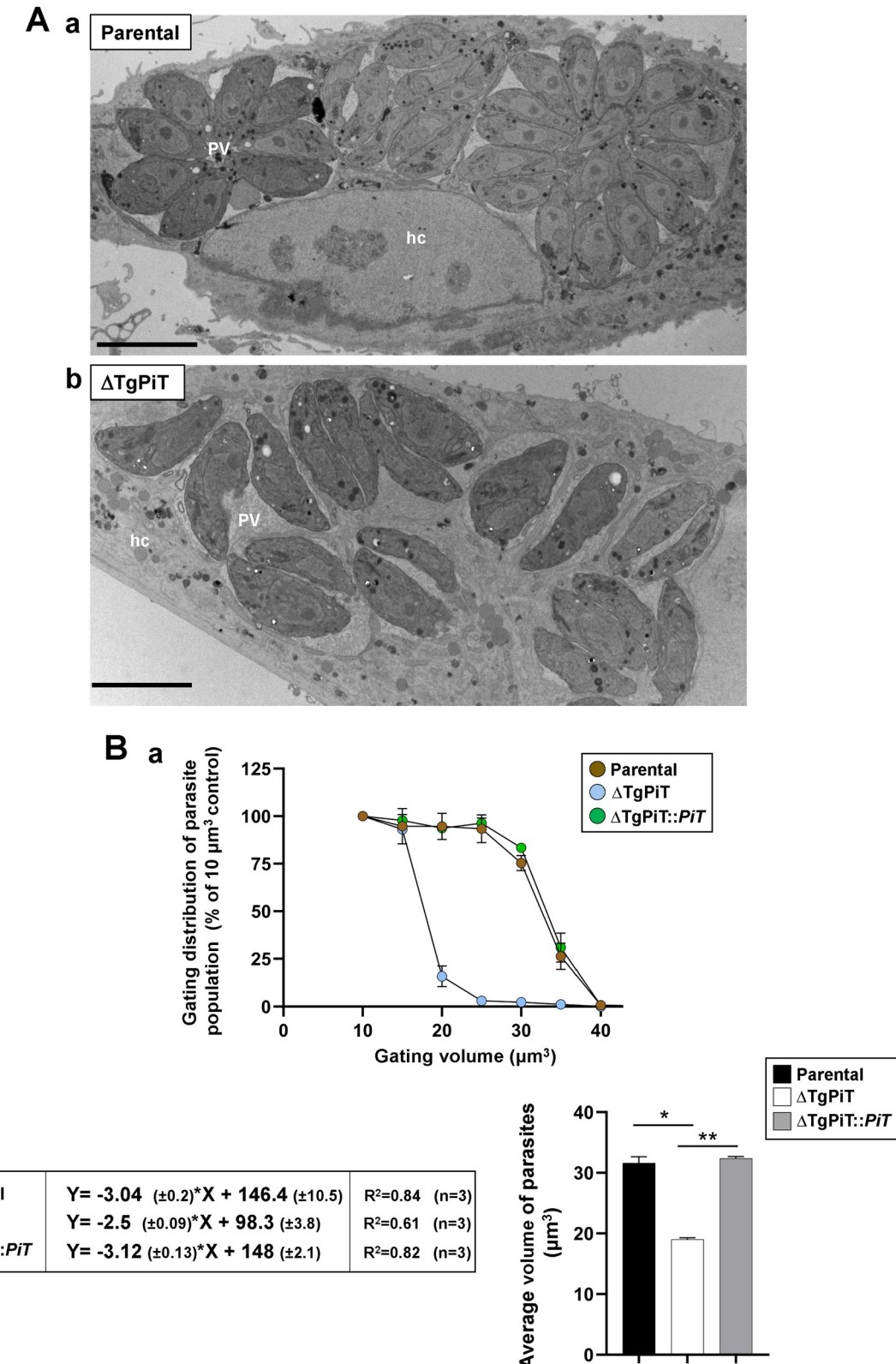

**Fig 7. Cell volume of ΔTgPiT parasites. A.** EM of HFF infected for 24 h with parental or ΔTgPiT parasites. Comparison between parental (panel a) and ΔTgPiT (panel b) parasites for PV size and parasite global morphology, showing abnormally thinner knockout parasites. Bars, 10 μm. **B.** Cell volume measurement of parental, ΔTgPiT and ΔTgPiT::*PiT* parasites using a Coulter counter. Panel a:

volume distribution of parasites. Extracellular parasites were counted after gating on the basis of 6 different volumes, starting from 10 μm³ to 40 μm³, with 5 μm³ increments. The percent of parasite population was normalized to 100% of the population at the 10 μm3 gate as all parasites were found to have a larger volume that this gate. Data are mean ± SEM, n = 3 independent assays with samples in triplicate. *P* values calculated using Fisher's LSD test were statistically significant between parental and ΔTgPiT parasites (*p* = 0.0173), and between complemented and ΔTgPiT parasites (*p* = 0.0023). Panel b: Linear regression equations for volume assessment of parental, ΔTgPiT and ΔTgPiT::*PiT* parasites. Average volumes were determined based on curves of the % population decline with increased gating volumes (in panel a) from each independent biological replicate. A linear regression was calculated with Y = % of parasite population and X = parasite volume. Data are mean ± SEM (n = 3). *, *p* = 0.0035; **, *p* < 0.0001 (Uncorrected Fisher's LSD).

in the mutant, suggesting a phosphate sensing mechanism that regulates the mobilization of phosphate-gathering acidocalcisomes.

## Upon phosphate restriction, *Toxoplasma* develops poorly

These results led us to investigate the level of dependence of *Toxoplasma* on exogenous phosphate. After 24 h of incubation of parental parasites in normal or low phosphate-containing medium, there was no significant difference in the number of parasites per PV (Fig 10A; black vs. magenta histograms). The numbers of ΔTgPiT parasites per PV were also not statistically different between mutant grown under normal or low phosphate growth conditions (white vs. green histograms). However, at low phosphate concentration, there was significant differences in PV size between parental and ΔTgPiT parasites, with the majority of the PV containing only 2 parasites for the mutant strain (green vs. magenta histograms), and these differences were more pronounced than those between parental and ΔTgPiT parasites incubated under normal phosphate condition (black vs. white histograms). This suggests a very high sensitivity of the knockout strain upon phosphate restriction.

When the exposure to low phosphate concentration was prolonged for 7 days, both parental and ΔTgPiT parasites showed significant growth delays, as observed on plaque assays (Fig 10B, panel a). Data quantification shows a ~40% and ~55% reduction in plaque number and size, respectively, for parental parasites cultivated at low phosphate concentration compared to normal phosphate conditions (Fig 9B, panel b). For knockout parasites maintained at low phosphate condition, a more severe growth defect was noticed, with a ~60% and ~65% reduction in plaque number and size, respectively, compared to normal conditions. Jointly, these observations indicate that *T*. *gondii* could adapt to low phosphate concentration for 2 to 3 cycles of replication (24 h of cultivation), for example by exploiting internal sources of phosphates, but for longer periods of time, exogenous sources of phosphate are required for normal development. The more dramatic growth impairment of the ΔTgPiT parasites maintained for days at low phosphate could be likely due to the combined effect of TgPiT loss and low phosphate cultivation. This would further diminish phosphate availability and the effectiveness of any efficient compensatory phosphate scavenging mechanisms when exogenous phosphate uptake is compromised.

We next performed EM studies to analyze the ultrastructure of ΔTgPiT and parental parasites upon $P_i$ deficiency during 24 h. Parental parasites formed large rosettes and did not show any abnormal features when grown at < 10 μM $P_i$ (Fig 10C, panel a) while ΔTgPiT parasites were skinner, with irregular body shapes (Fig 10C, panel b). Some knockout parasites also exhibited replication defects, with disorganized endodyogeny (Fig 10D, white arrowheads in panel a) or aborted progeny (Fig 10D, green arrowheads in panel b), and thus very low number of parasites per PV. The PV lumen were often filled with membranous debris (Fig 10C, panel b and Fig 10D, panel b, asterisks), which could be remnants of dead parasites. Finally, many profiles of acidocalcisomes were observed in the cytoplasm of ΔTgPiT parasites (Fig 10D, panel c). These observations confirm that under conditions of poor environmental $P_i$ availability,

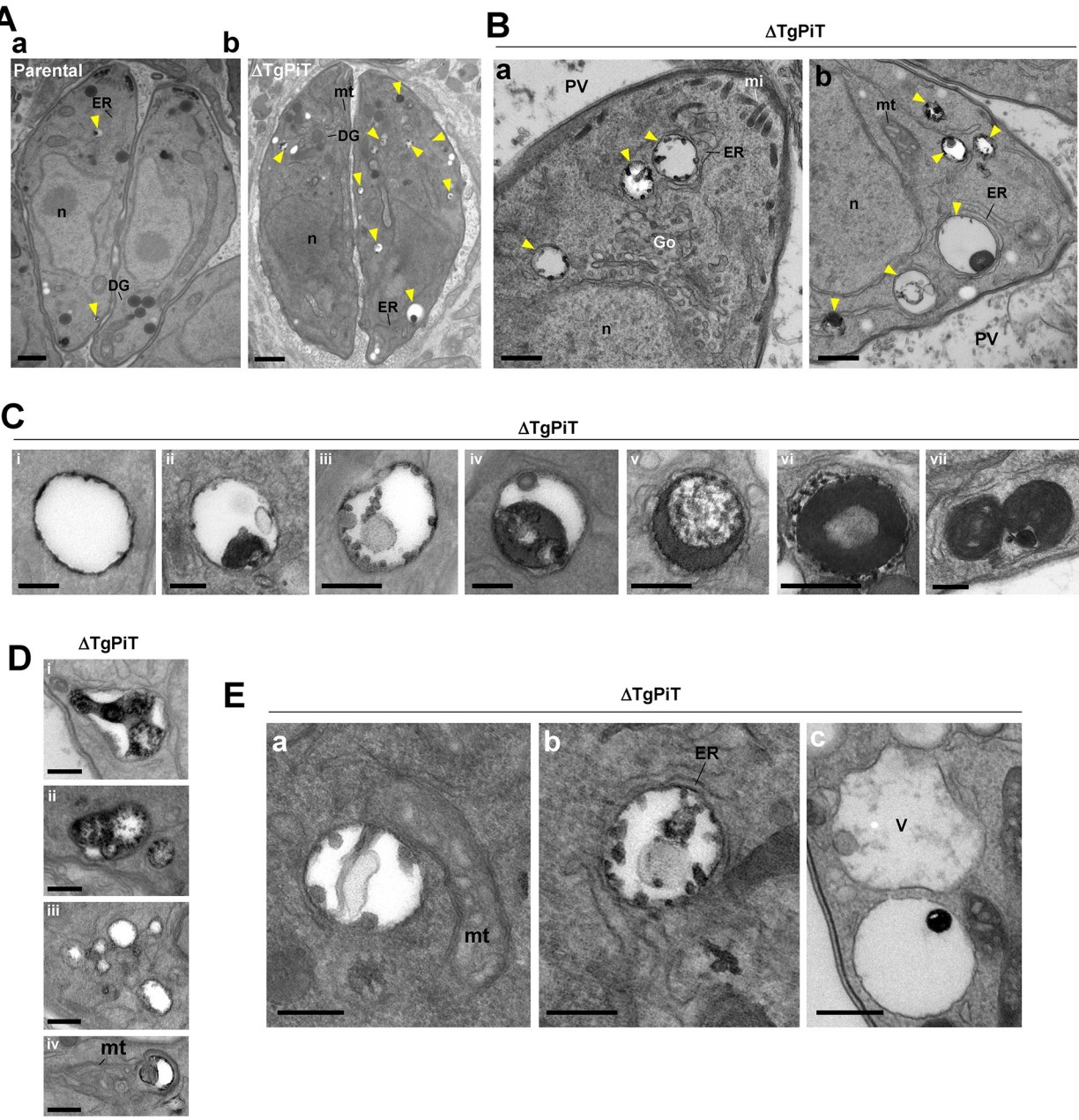

**Fig 8. Acidocalcisomes in ΔTgPiT parasites. A.** EM of HFF infected for 24 h with parental or ΔTgPiT parasites. Comparison between parental (panel a) and ΔTgPiT (panel b) parasites for acidocalcisome (arrowheads) content. Bars, 500 nm. **B.** Ultrastructure of acidocalcisomes (arrowheads) in ΔTgPiT parasites, typified by luminal electron-dense inclusions. Bars, 500 nm. **C-E.** Panel of different acidocalcisomes in ΔTgPiT parasites, showing: in D from panel i to vii increased electron-dense material in the matrix; in E various shape and; in F proximity to other organelles. Bars in D-F, 200 nm. DG, dense granule; Go, Golgi; hc, host cell; m, mitochondrion; mi, microneme; n, nucleus; V, the VAC.

knockout parasites could not properly adapt, even for a short period of 24 h, and could barely divide.

## The VAC compartments are enlarged in ΔTgPiT parasites

TgPiT localizes to the inward buds of the outer membrane of the VAC compartments. This prompted us to examine the morphology of VAC in the absence of PiT expression.

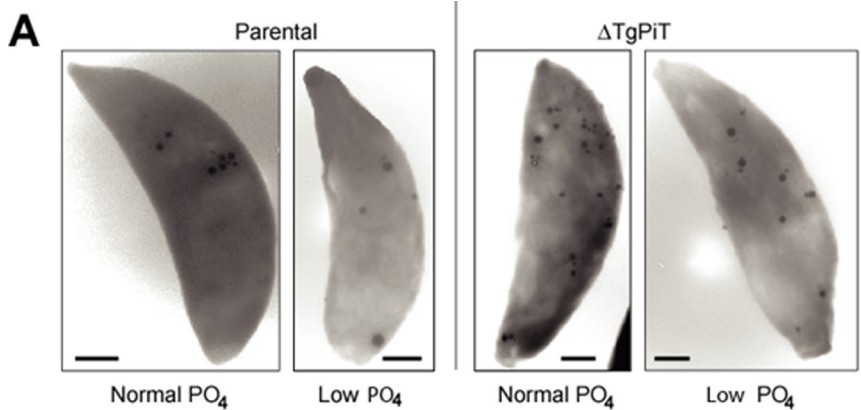

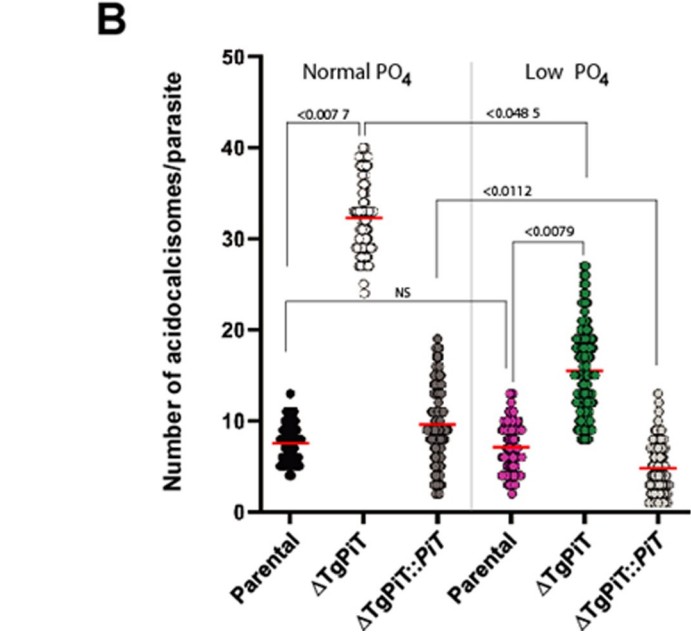

**Fig 9. Quantification of acidocalcisomes in ΔTgPiT parasites. A.** EM of representative extracellular parental and ΔTgPiT parasites incubated in normal or low PO$_4$ medium for 24 h and applied to carbon-coated formvar grids before examination at the microscope, showing the abundance of acidocalcisomes in the mutant. **B.** Dotplot graphs for acidocalcisome number per parasite strain per PO$_4$ condition (50 parasites observed in 3 independent preparations). *p* values were calculated using Fisher's LSD test.

Intracellular parental and ΔTgPiT parasites 24 h p.i. were immunostained with antibodies against cathepsin L (CPL), a major luminal protease in the VAC [39]. Compared to parental parasites, ΔTgPiT parasites showed a brighter and more pronounced signal for CPL (Fig 11A). In extracellular, egressing parasites, VAC is one large apical organelle; following invasion, VAC fragments into smaller vesicles proportionally to the number of replication cycles. We next inspected PV containing the same number of parasites for a direct comparison between the parasite strains and confirmed that the CPL-containing structures in ΔTgPiT parasites were enlarged or swollen, as compared to parental parasites.

We then examined the ultrastructure of VAC in extracellular parasites in which VAC is a prominent feature. EM observations show that the VAC appeared abnormally dilated in

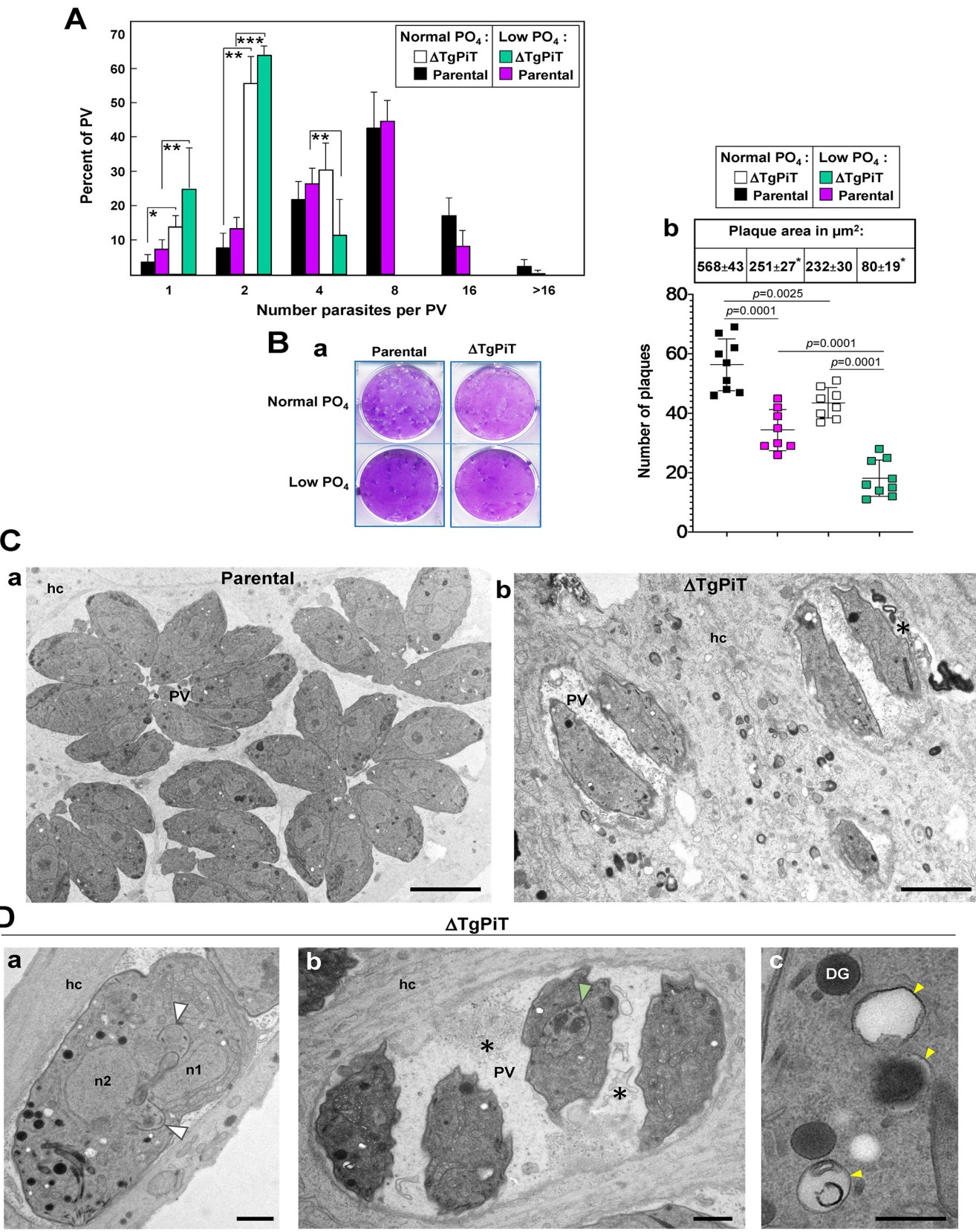

**Fig 10. Development and ultrastructure of ΔTgPiT parasites upon phosphate deprivation. A.** Quantitative measurement of parental and ΔTgPiT parasite replication 24 h p.i. assessed by parasite enumeration after incubation in normal or low $PO_4$ conditions. No statistical differences using a *p*-value are observed between normal and low $PO_4$ concentrations within each parasite group. Statistical differences are observed between parental or ΔTgPiT parasites at normal and low $PO_4$ concentrations. *, $p < 0.05$; **, $p < 0.001$; ***, $p < 0.0025$ (unpaired Student's *t*-test). **B.** Parasite growth quantification by plaque assays. Confluent monolayers of HFF were infected with 100 parental or ΔTgPiT parasites and maintained in normal or low $PO_4$ for 7 days before counting the plaques and measuring their size, from 4 or 5 independent assays in triplicate. Panel a shows representative images of lysis plaques for the 4 conditions. Panel b are dotplot graphs for plaque number with *p*-values (unpaired Student's *t*-test) and the table is means ± SD of plaque area (*, $p < 0.01$ between normal and low $PO_4$ within each parasite group). **C-D.** EM of HFF infected for 24 h with parental or ΔTgPiT parasites at low $PO_4$. **C.** Comparison between parental (panel a) and ΔTgPiT (panel b) parasites for PV size and parasite global morphology, showing rachitic knockout parasites. Bars, 5 μm. **D.** ΔTgPiT parasites showing: in panel a, an abnormally enlarged dividing parasite with two nuclei (n1 and n2), two nascent apexes (arrowheads), in panel b, four misshapen parasites with one poorly dividing (arrowhead) and membranar debris (asterisks as in Fig 11A, panel b) in the PV lumen, and in panel c, 3 acidocalcisomes (arrowheads). Bars, 500 nm. DG, dense granules; hc, host cell; n, nucleus.

ΔTgPiT parasites, compared to parental parasites (Fig 11B, panel a). The total VAC surface area for the ΔTgPiT parasites was $6.33 \pm 0.58$ μm$^2$, approximately 5-fold larger than that of VAC in parental parasites corresponding to $1.26 \pm 0.25$ μm$^2$ (Fig 11B, panel b). We also inspected the content and the shape of VAC compartments in ΔTgPiT parasites compared to parental parasites by EM. In extracellular parental/WT parasites, VAC organelles were generally spherical, but sometimes had an angular (triangular or square) shape, and they contained few materials (not shown). By contrast, up to 36% of VAC in ΔTgPiT parasites contained a lot of e-dense materials, including membranous structures and vesicles (Fig 11C, panels a and b). Of interest, 18% of VAC had dramatic various morphologies, such as a star-like shape with multiple indentations (Fig 11C, panel c), suggestive of an unusual remodeling of the limiting membrane of VAC. Several acidocalcisomes were seen closely apposed to VAC structures (Fig 11D), suggestive of membrane contacts between the two organelles, perhaps for exchange of materials. Overall, these observations suggest that the loss of TgPiT on VAC alters VAC morphology, content, and likely its osmoregulatory function.

## TgPiT contributes to the regulation of intracellular pH and the release of calcium from acidic organelles

The reduced cell volume of ΔTgPiT parasites suggest dysfunctions in water balance and osmoregulation. This phenotype is possibly linked to changes in the ionic status in the mutant, due to its impaired ability to mobilize phosphate and sodium ions. Supporting this hypothesis, it has been reported that free phosphate and calcium concentrations are inextricably linked, and specifically to *Toxoplasma*, calcium retention in the acidic compartments of *T. gondii* is directly attributable to the calcium chelating properties of polyP accumulated in acidocalcisomes [46–48]. Furthermore, sodium homeostasis and cytosolic proton load have been proposed to be interconnected in *T. gondii* and *Plasmodium* species [30, 31]. To explore the scope of ionic imbalance and provide insights on the cause of the observed osmoregulatory defect in ΔTgPiT parasites, we measured the cytosolic $Ca^{2+}$ and $H^+$ levels in the mutant in comparison to parental and complemented parasites. To investigate the potential role of TgPiT in $Ca^{2+}$ regulation and storage, we used the ratiometric $Ca^{2+}$-sensitive dye Fura-4F-AM to measure the cytosolic concentration of calcium. The basal levels of $Ca^{2+}$ were not significantly different between the 3 strains (Fig 12A, panels a and b). The addition of $NH_4Cl$ triggers the release of calcium from acidic organelles, which indictaes the amount of calcium retained in acidic compartments [49]. Compared to control parasites, ΔTgPiT parasites were significantly less responsive to $NH_4Cl$-induced $Ca^{2+}$ release from acidic compartments (e.g., VAC and acidocalcisomes) into the cytosol. This suggests that the mutant has defects in $Ca^{2+}$ storage and/or $Ca^{2+}$ demobilization from its acidic compartments.

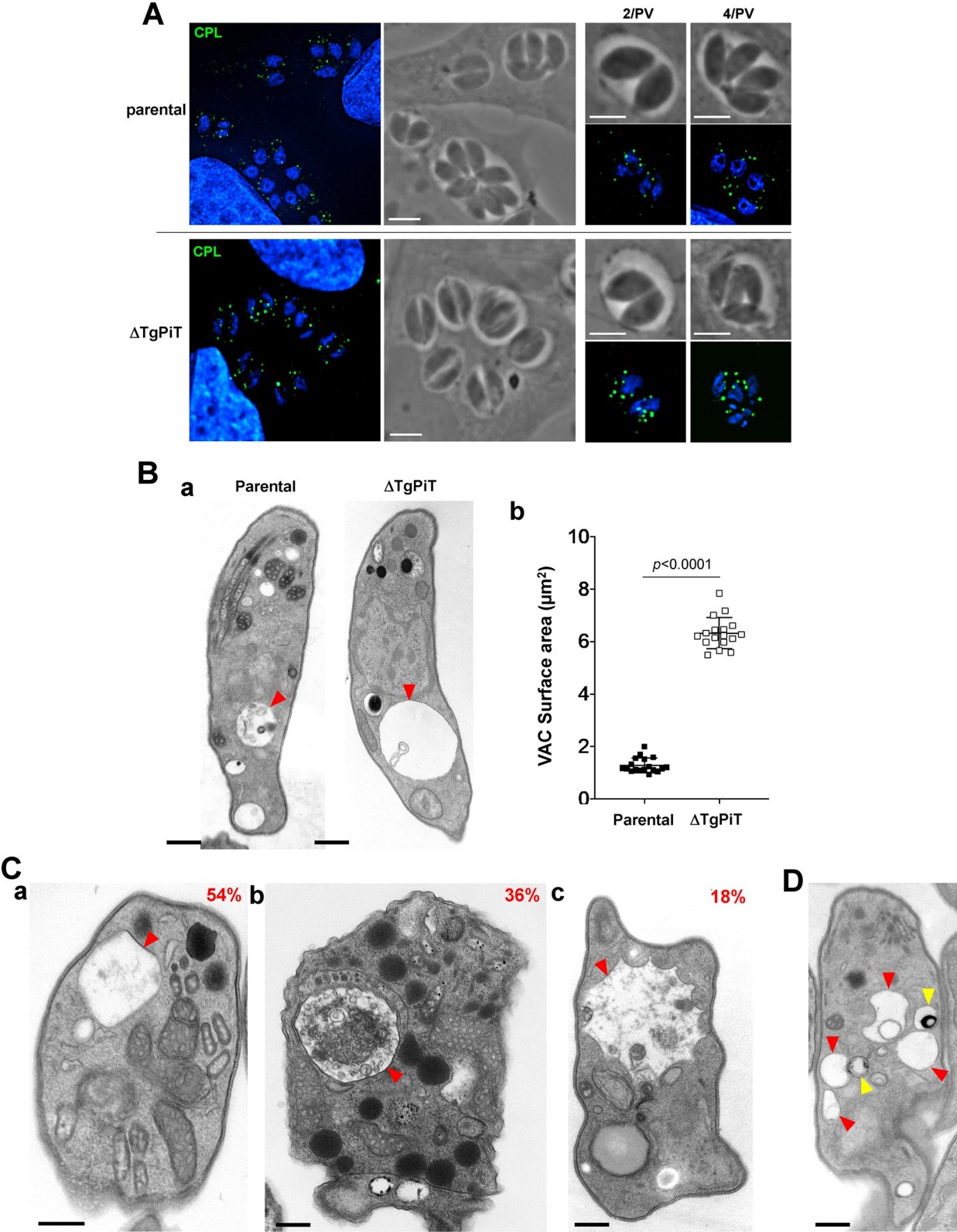

**Fig 11. Ultrastructure of VAC in ΔTgPiT parasites. A.** IFA on HFF infected for 24 h with parental or ΔTgPiT parasites stained with anti-CPL antibodies showing a strong signal for the mutant. Scale bars, 5 μm. **B.** Panel a: EM of extracellular parental or ΔTgPiT parasites collected during their egress from HFF prior to fixation, comparing VAC (arrowheads) size at the same magnification on these representative images. Bars, 500

nm. Panel b: Measurement of the VAC surface area from 18 independent electron micrographs of parasites from each group, showing a significant increase in VAC size in parasites lacking TgPiT, compared to the parental strain. Data are means ± SD. Statistical significance was determined using unpaired Student's *t*-test. **C.** Representative EM of VAC (red arrowheads) from ΔTgPiT parasites, characterized by e-lucent content (panel a), luminal accumulation of material (panel b), or irregular shape (panel c), with the % for each phenotypes from 55 VAC. Bars, 300 nm. **D.** EM of a ΔTgPiT parasite showing VAC-acidocalcisome interactions. VAC, red arrowheads; acidocalcisomes, yellow arrowheads. Bars, 300 nm.

We next measured the intracellular pH in parasites from the 3 strains after loading the parasites with the ratiometric pH-sensitive fluorescent indicator BCECF-AM. The average resting pH in parental and complemented parasites was 7.392 and 7.386, respectively, but

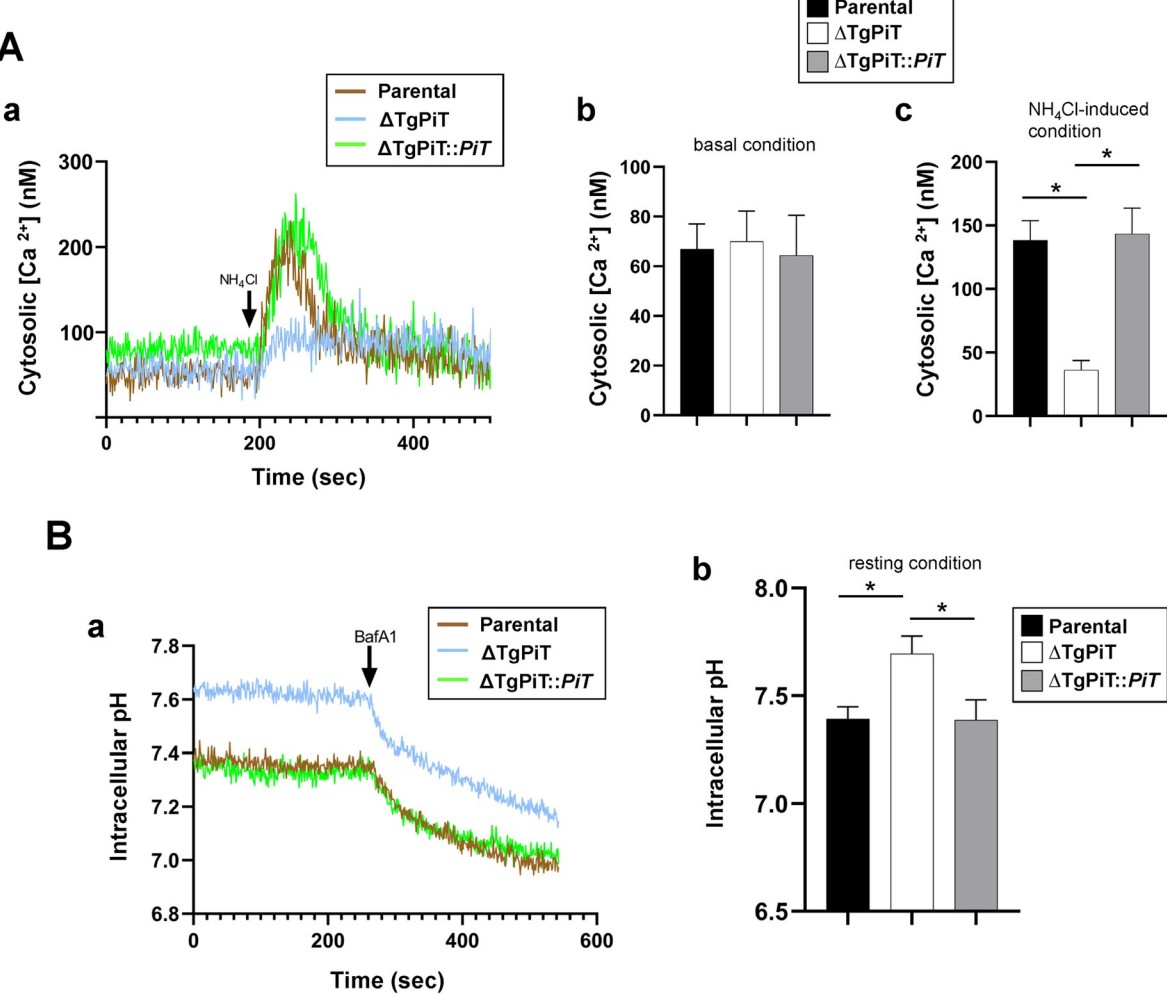

**Fig 12. Ion homeostasis in ΔTgPiT parasites. A.** Calcium regulation in ΔTgPiT parasites and the effect of NH₄Cl-induced calcium release from acidic compartments. Panel a: Representative tracings of cytosolic Ca²⁺ measurements on extracellular parental, ΔTgPiT and ΔTgPiT::*PiT* parasites loaded with the ratiometric fluorescent indicator Fura-4F-AM. Each tracing is representative of 3 independent experiments. NH₄Cl at 10 mM was added at 200 sec to induce Ca⁺⁺ release from acidic stores. Panels b and c: Quantification of tracings from 3 independent experiments showing the ratiometric Fura-4F-AM fluorescence for the 3 parasite strains, before and after NH₄Cl addition. Data are mean ± SEM. *, $p = 0.0078$ (KO vs. parental) and $p = 0.0061$ (KO vs. complemented) (Tukey's multiple comparisons test). **B.** pH regulation in ΔTgPiT parasites and the effect of bafilomycin A1 (BafA1). Representative tracings showing the intracellular pH of parental, ΔTgPiT and ΔTgPiT::*PiT* parasites measured via BCECF-AM ratiometric fluorescence. Each tracing is representative of 3 independent experiments. The V-H+-ATPase inhibitor BafA1 at 10 nM was added at 250 sec to depolarize the plasma membrane. Panel b: Quantification of tracings from 3 independent experiments showing the average intracellular pH in the three parasite strains Data are mean ± SEM. *, $p<0.0001$ (Tukey's multiple comparisons test).

corresponded to 7.694 in the knockout, indicating mild alkalinization of the cytosol in ΔTgPiT parasites (Fig 12B, panels a and b). However, addition of the membrane depolarizer bafilomycin A1 to the parasite samples resulted in a similar drop in pH due to proton retention in the cytosol for the 3 strains. This suggests that the mutant is able to reverse the alkalinization of the cytosol upon inhibition of V-H$^+$-ATPase. In conclusion, these data reveal that TgPiT is involved in maintaining storage and/or accessibility of Ca$^{2+}$ from acidic stores and in regulating the intracellular concentration of H$^+$, thus stabilizing the cytosolic pH.

## ΔTgPiT parasites have altered gene transcription to assist in the adaptation of P$_i$ uptake and storage

We performed RNA-Seq analysis on ΔTgPiT versus parental parasites to identify possible genes with altered transcription levels, in relation to reduced P$_i$ import and storage, stimulation of acidocalcisome biogenesis and control of VAC size. A large number of genes have significantly different expression levels in ΔTgPiT parasites, with 281 genes that are up-regulated and 162 genes down-regulated in the knockout strain versus parental parasites (Fig 13A). Among the genes that have increased or reduced transcripts, 42% and 76% could be identified in the *Toxoplasma* database (www.ToxoDB.org) or blasted in Swissprot database, respectively (Fig 13B and 13C, see details in S2–S5 Tables).

Loss of TgPiT results in the up-regulation of several transporters/carriers, although their transferred ligands remain to be identified, e.g., carrier superfamily protein TGGT1_235650; Log$_2$ fold change (logFC): 1.6), two Major Facilitator Superfamily proteins (TGGT1_216710; LogFC: 1.6 and TGGT1_266870; LogFC: 0.9), and two ABC transporters (TGGT1_239020; LogFC: 0.7 and TGGT1_263740; LogFC: 0.5). Interestingly, the gene product of TGGT1_216710 shares 27% identity with the P$_i$:H$^+$ symporter (PHS) member: H$^+$-linked myoinositol transporter from *Trypanosoma brucei* (TbHMIT) [50].

ΔTgPiT parasites have up-regulated several genes encoding enzymes that cleave or transfer phosphate groups from molecules to others, such as NTPase (TGGT1_225290; LogFC: 3), Ca$^{2+}$-ATPase (TGGT1_288520; LogFC: 1.8), acid phosphatase (TGGT1_228160; LogFC: 1.5), and several kinases, e.g. adenylate kinase (TGGT1_269050; LogFC: 1.5), protein kinase (TGGT1_236620; LogFC: 0.7), suggesting an adaptation of the knockout to the critical situation of dramatically reduced P$_i$ internalization. These enzymes might contribute to regulate intracellular P$_i$ resources and parasite survival in the absence of an exogenous supply of P$_i$.

Many genes encoding for bradyzoites markers, such as surface proteins and glycolytic enzymes selectively expressed by *T. gondii* cyst forms, are also up-regulated in ΔTgPiT parasites. This suggests that a limiting P$_i$ supply could represent a stressful condition for *Toxoplasma*, which is on the verge of switching from tachyzoites to slow growing latent bradyzoites, as previously demonstrated when the parasite is exposed to starvation conditions [51].

In accordance with the abundance of acidocalcisomes and the increased size of VAC compartments in ΔTgPiT parasites, genes coding for transporters or ions exchangers localized to these organelles are up-regulated, such as a proton-type ATPase that forms gradients across membranes by translocating cations, heavy metals and lipids (TGGT1_257720; LogFC: 4.3), P-type ATPase PMA1 (TGGT1_252640; LogFC: 3.1), and voltage-dependent N-type calcium channel for cation transport (TGGT1_205265; LogFC: 1.1). Of note, despite the absence of canonical peroxisomes in *Toxoplasma*, a gene coding a peroxisomal biogenesis factor PEX11 is one of the highest up-regulated genes in ΔTgPiT parasites (TGGT1_243720; LogFC: 4.9). The PEX11 family of homologous proteins is responsible for the proliferation of peroxisomes, functioning as a pore-forming protein sharing sequence similarity with transient receptor

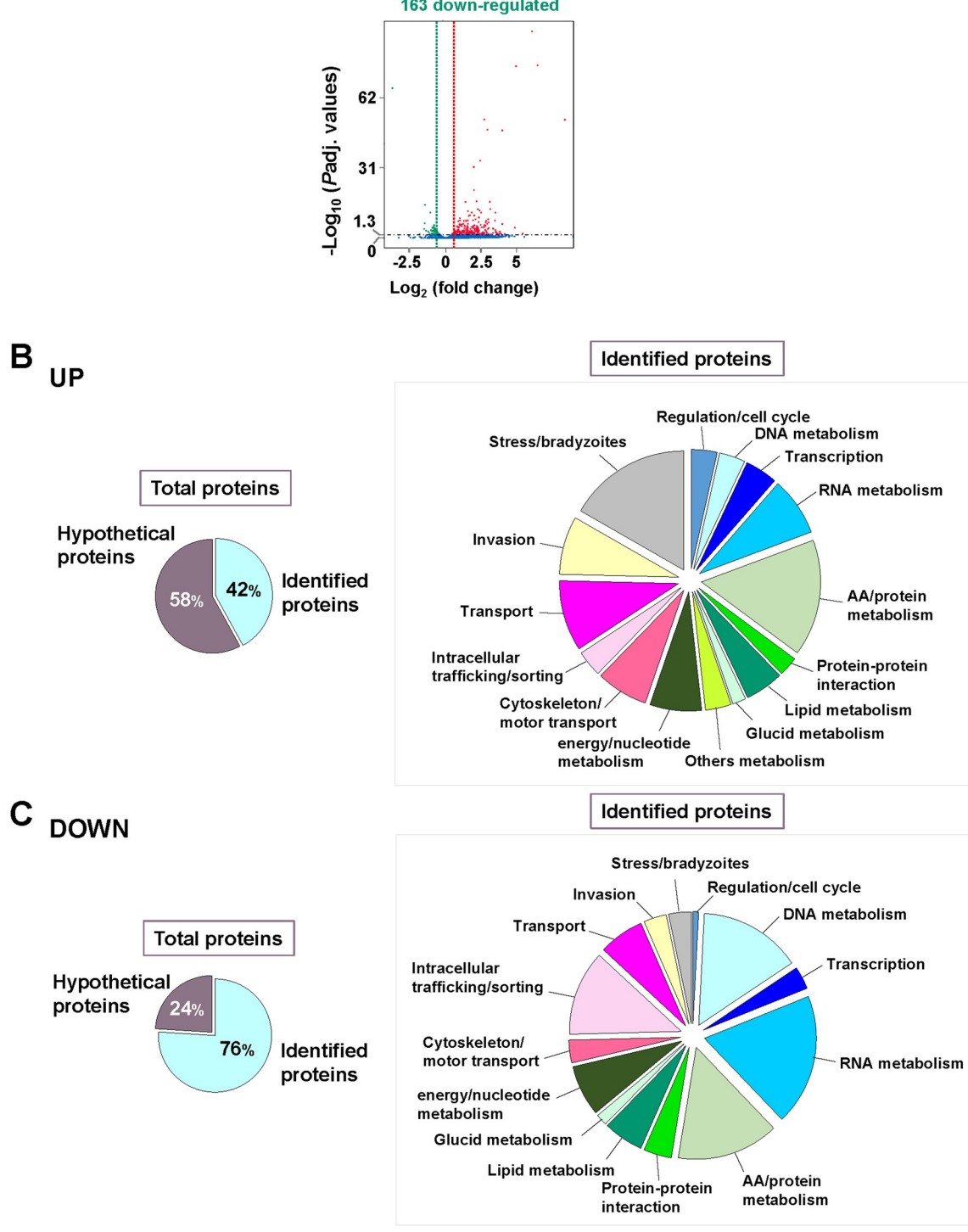

**Fig 13. RNA-Seq comparison between ΔTgPiT and parental parasites. A.** Volcano plot revealing 281 genes having increased expression (red) and 163 with decreased expression (green), with statistical significance less than 0.05 in the ΔTgPiT relative to the parental strain. The green and red dashed lines represent the borderline of $Log_2$ fold change of 0.4 in gene transcripts, and the genes above the black dashed line had padj values of statistical significance below 0.05. Each sample was sequenced in duplicate for statistical comparison. **B-C.** Left: Pie charts plotting the percent of proteins whose genes are up-regulated (in B) or down-regulated (in C), identified in the ToxoDB. Right: Pie charts categorizing the identified proteins based on their biological functions.

potential (TRP) cation-selective channels [52]. This suggests that if PEX11 is expressed on acidocalcisomes or VAC, it may be involved in the translocation of monovalent ($K^+$/$Na^+$) and/or divalent ($Ca^{2+}$) cations across the membrane of these organelles. A gene encoding a sulfate permease (TGGT1_280500-t26; LogFC: 3.5) is highly up-regulated in the knockout, suggesting perhaps a coordination of phosphate and sulfate homeostasis in *Toxoplasma*.

The morphology of VAC compartments resembles that of multivesicular bodies (MVB). In ΔTgPiT parasites, a gene coding for a SNF7 family protein (TGGT1_220460; LogFCg: 1.7), which is part of the ESCRT-III complex and plays a role in endosome-mediated trafficking via MVB formation and sorting, is up-regulated. The *Toxoplasma* SNF7 homolog may contribute to the transport of proteins to the VAC. Of note, the three genes that are the most up-regulated from LogFC 8.4 to 6.1 (TGGT1_216290A, TGGT1_216290B and TGGT1_216335) code for hypothetical proteins containing a Vacuolar Protein Sorting (VPS) motif domain and may be involved in intracellular transport and vesicle-mediated sorting for VAC compartments and/or acidocalcisomes.

ΔTgPiT parasites likely undergo osmotic imbalance as evidenced by the enlarged, swollen VAC compartments and shrunken cell body. In relation to these phenotypic features, a class of proteins whose genes are up-regulated are proteolytic enzymes including a membrane alanyla-minopeptidase N (TGGT1_221310; LogFC: 5), two subtilisins (SUB8: TGGT1_235950; LogFC: 2.9 and SUB3: TGGT1_200350; LogFC: 0.9), aspartyl protease ASP1 (TGGT1_201840; LogFC: 1.7), CocE/NonD family protein (TGGT1_227370; LogFC: 1.75), kazal-type serine protease inhibitor 2 (TGGT1_208450; LogFC: 0.7) and several proteins involved in proteasome function. By degrading proteins, these enzymes may increase the intracellular pool of pep-tides/amino acids and may act as compatible osmolytes to replace the inorganic ions ($Na^+$, $K^+$) sequestered in polypophosphates in the expanded population of acidocalcisomes in the knock-out, and thus prevent cellular damage from reducing increased cytosolic ionic strength. In addition, the expression of a gene encoding a small conductance mechanosensitive ion chan-nel (TGGT1_219650; LogFC: 1.1) involved in the regulation of osmotic pressure changes in response to stretch forces in the membrane lipid bilayer, is also increased.

The expansion in number and size of acidocalcisomes and VAC compartments would require large amounts of lipids to form these organelles. Few genes encoding for enzymes involved in lipid metabolism are up-regulated, such as a sterol-sensing domain of SREBP cleavage-activation domain-containing protein (TGGT1_295020; LogFC: 2).

Of interest, genes encoding enzymes involved in unexpected metabolites are up-regulated in ΔTgPiT parasites, such as a radical SAM-domain containing protein (TGGT1_288640; LogFC: 1.5) putatively involved in porphyrin metabolism, MoeA N-terminal region domain-containing protein (TGGT1_293480; LogFC: 1.9) and a molybdopterin converting factor (TGGT1_273350; LogFC: 0.9) producing molybdopterin, 8-amino-7-oxononanoate synthase (TGGT1_290970; LogFC: 2.4) for biotin synthesis, a putative raffinose synthase 1 (TGGT1_283810; LogFC: 0.6) and a Vitamin K epoxide reductase family protein (TGGT1_203720; LogFC: 1.7). It could be interesting to determine whether these metabolites are produced *in situ* in acidocalcisomes or VAC compartments, important for the function of these organelles or destined for storage in these organelles.

Many different genes involved in mRNA and tRNA tasks, and coding for AP2 domain tran-scription factors have altered expression in ΔTgPiT parasites, with more down-regulation than up-regulation, reflecting dramatic changes in metabolic pathways in the knockout. The most down-regulated gene encodes a 3′5′-cyclic nucleotide phosphodiesterase (PDEase) domain-containing protein (TGGT1_226755; LogFC: -3.8) involved in signal transduction. Of note, a gene coding for another PDEase (TGGT1_220420; LogFC: 1.7) is up-regulated in the knock-out, likely involved in another signaling network.

ΔTgPiT parasites down-regulate the expression of the apicoplast phosphate translocator APT1 (TGGT1_261070; LogFC: -0.8) that imports $P_i$ required for fatty acid synthesis in the apicoplast [53, 54] and that of a succinyl coenzyme A synthetase that facilitates the formation of GTP or ATP from GDP or ADP and $P_i$, perhaps as another strategy to redistribute the intracellular resources of $P_i$ when $P_i$ import is limited. The mutant also decreases the expression of a gene coding for a sulfate permease family protein (TGGT1_287230; LogFC: -0.7) that is different from the one that is up-regulated. The differential expression of these permeases, whether located to different membrane or involved in translocating sulfate in opposite directions, suggests that the regulation of sulfate transport may be part of the parasite adaptation to the condition of $P_i$ starvation.

Finally, four genes encoding a CorA family $Mg^{2+}$ transporter protein (TGGT1_273970; LogFC: -0.9), a vacuolar ATPase subunit C (TGGT1_315620; LogFC: -0.7) hydrolyzing ATP to catalyze the transmembrane movement of $H^+$ ions, a large neutral amino acids transporter (TGGT1_263260; Log: -0.6) and an ABCG transporter (TGGT1_305590; LogFC: -0.6) facilitating the translocation of lipophilic molecules, are also down-regulated in the knockout. Although the localization of these transporters and directional transport of their ligand are still unknown, it suggests several alterations in trafficking pathways in ΔTgPiT parasites.

## Discussion

Active translocation of $P_i$ across the plasma membrane is central for the maintenance of $P_i$ homeostasis and is an initial step in the exploitation of this essential anion, which is involved in many cellular functions. Low availability of $P_i$ in the environment is a limiting factor for the growth of several organisms, including unicellular eukaryotes [10]. Here, we demonstrate the dependence of intracellular *Toxoplasma* on exogenous $P_i$ for its optimal growth. The uptake of $P_i$ by the parasite is coupled to an inwardly directed $Na^+$ gradient that facilitates the import of $P_i$ against a concentration gradient. The parasite expresses a unique TgPiT, a selective $P_i$ transporter that localizes at the plasma membrane and on VAC internal membranes. Finally, by selective gene deletion and complementation in the mutant, we ascertain that TgPiT contributes to parasite survival, virulence, $Na^+$ and $Ca^{2+}$ homeostasis, neutral pH maintenance and osmoregulation.

Among the sources of exogenous $P_i$ available to *Toxoplasma*, free $P_i$ in the host cytosol can penetrate into the PV through pores within the PV membrane [11] and be readily available to the parasite. An alternative source of $P_i$ for intravacuolar *Toxoplasma* may be phosphorylated molecules in the PV lumen. In this case, the parasite would need to liberate $P_i$ from substrates such as ATP or other phosphate-bound molecules, by secreting phosphatase-like enzymes into the PV. Some protozoan parasites (e.g., *Plasmodium* or Trypanosomatidae) secrete phosphatases (termed purple acid phosphatase or secreted acid phosphatase) to dephosphorylate nutrients [55, 56]. In response to phosphate limitation in the medium, Trypanosomatidae also express ectophosphatases at the plasma membrane, with the catalytic site facing the external medium, as a compensatory mechanism used to increase the supply of $P_i$ [57–59]. Despite having a common enzymatic function, acid phosphatases (EC 3.1.3.2) differ with respect to amino acid homology, metal dependency and resistance to tartrate. Several phosphatases have been annotated in the *T. gondii* genome database although the majority of them belong to a family of enzymatically inactive protein phosphatases [60]. However, we identified a protein annotated as a Ser/Thr phosphatase (TGME49_297650) with potential acid phosphatase activity, and a signal peptide predictive of its secretion. This Ser/Thr phosphatase possesses significant homology to purple acid phosphatases, especially to type 5 acid phosphatases. Interestingly, type 5 acid phosphatases in plants are involved in $P_i$ scavenging or recycling under conditions

of low phosphorus availability [61]. The *Toxoplasma* Ser/Thr phosphatase shares 27% and 32% identity with the human homologue AcP5b (AC #: P13686) and the plant homologue *A. thaliana* AtACP5 (AC #: AJ133747), respectively. Similar to purple acid phosphatases, the parasite enzyme contains a dinuclear metal center, and includes the conserved residues involved in metal ligand binding and resistance to tartrate inhibition. Thus, this potential Ser/Thr phosphatase may function to provide $P_i$ to intravacuolar *Toxoplasma*.

TgPiT belongs to a class of high-affinity $P_i$ transporters that are characterized by $K_m$ values ranging from 5 to 150 μM for $P_i$ binding. The $K_m$ value of 23 μM for $P_i$ uptake by *Toxoplasma* is similar to those observed for other protozoan parasites, such as *Trypanosoma cruzi*, *Trypanosoma rangeli* and *Leishmania amazonensis* [62–64], but is lower than calculated for *P. falciparum* with a $K_m$ value of 106 μM [6]. Both TgPiT and PfPiT have a preference for $H_2PO_4^-$ (over other phosphate ions) that is co-internalized with $2Na^+$, resulting in a net influx of a +1 charge in these parasites. TgPiT- and PfPiT-mediated $P_i$ transport are thus both driven by a $Na^+$ gradient that provides the driving force for $P_i$ uptake. The high cooperativity (Hill) index ($n_H = 4.6$) of TgPiT suggests the juxtaposition of negatively charged residues on TgPiT acting as high-affinity $Na^+$ binding sites, with at least 2 $Na^+$ ions to promote the binding of one $P_i$ molecule. In unicellular eukaryotes, members of the PiT family contain 354 to 681 amino acid residues and 10 to 12 transmembrane domains [65], making TgPiT a larger transporter than other PiT members. The localization of TgPiT to intracellular compartments, in addition to the plasma membrane is unusual and implies that the transporter functions to move $P_i$ across organellar membranes. The TgPiT's capacity to mobilize both extracellular and intracellular pools of $P_i$, portrays the expansive role of this transporter in phosphate homeostasis in *Toxoplasma*.

The availability of nutrients often limits the growth of microbes. In many organisms, high-affinity transport systems for nutrients like phosphate, are transcriptionally up-regulated in response to nutrient limitation while some low-affinity nutrient transporters are selectively down-regulated, likely for energy preservation [66, 67]. In response to low phosphate conditions, *Toxoplasma* does not up-regulate nor down-regulate TgPiT expression, suggesting constitutive expression. Nevertheless, the parasite has the ability to sense and respond to changes in phosphate availability through the delocalization of TgPiT from intracellular pools to the plasma membrane. Our experimental setting exposing the parasite to a low phosphate condition loosely replicates the changing ionic environment that the parasite faces during its lifecycle, shuttling from the high phosphate intracellular environment to the low phosphate extracellular environment. One could speculate that TgPiT delocalization reflects a remarkable adaptation of the parasite for phosphate exploitation: from acquisition, storage, and use in metabolite and lipid synthesis during its intracellular replication, to mobilization for invasion and signaling processes when present in the extracellular environment. The growth defects of *Toxoplasma* mutant with reduced TgATP4 expression or activity, resulting in impaired $Na^+$ expulsion [31], may be attributed to toxicity from high levels of $Na^+$ in the cytosol. We show that TgPiT uses an extracellular $Na^+$ gradient to drive $P_i$ import into the parasite. Therefore, if TgATP4 is one of the major generators of concentration gradients of $Na^+$ ions outside the parasite, it may imply that TgATP4-deficient parasites also suffer from reduced $P_i$ internalization and availability. It would be interesting to determine the $P_i$ concentration in TgATP4-deficient parasites to verify this hypothesis which would further confirm the essentiality of $P_i$ as a nutrient for *Toxoplasma*.

In response to limited exogenous phosphate accessibility, ΔTgPiT parasites react through the transcriptional up-regulation of genes involved in phosphate demobilization from phosphate-containing molecules, or possibly internal phosphate stores (e.g., polyP) in acidocalcisomes. In unicellular eukaryotes, the second main $P_i$ transporter is the $P_i$:$H^+$ symporter (PHS)

(TC No. 2.A.1.9.1), which is included in the Major Facilitator Superfamily (MFS), such as yeast PHO84 [68], PHO-5 from *Neurospora crassa* [69] and TbHMIT from *Trypanosoma brucei* [50]. Interestingly, in ΔTgPiT parasites, a MFS protein (TGGT1_216710) is up-regulated, suggesting that it could function as a potential $H^+$:$P_i$ transporter at the plasma membrane. If this is the case, then this transporter may account for the uptake of $P_i$ in the absence of sodium (upon choline replacement) by *Toxoplasma*, as well as the uptake of trace amounts of $P_i$ by ΔTgPiT parasites, contributing to the survival of this mutant.

ΔTgPiT parasites have an intracellular pH that is slightly alkaline. ΔTgPiT parasites have an intracellular pH that is slightly alkaline. A link between pH balance and $Na^+$ homeostasis has been previously demonstrated for *T. gondii* and malaria parasites as the activity of the ATP4 ion pump is important not only to maintain resting cytosolic $Na^+$ concentration but also for $H^+$ import into the cytosol, since it exports $Na^+$ in exchange for $H^+$ [31, 70]. Subsequently, ATP4 imposes a significant acid load on the cytosol of these parasites. In the absence of TgPiT, the mutant may activate some compensatory mechanisms to adapt to its altered sodium acquisition profile and maintain the integrity of $Na^+$ homeostasis. From our data, we generated a model of $Na^+$ homeostasis in *T. gondii* that is achieved through a mosaic of transporters and enzymes at the plasma membrane (Fig 14). ATP4 represents the removal arm of $Na^+$ homeostasis while the uptake arm involves the $Na^+$-$H^+$ exchanger TgNHE1 that exchanges extracellular $Na^+$ for intracellular $H^+$, and TgPiT that co-imports $Na^+$ and $P_i$ into the cytosol. As with any organism, it is unlikely that these uptake and export activities for $Na^+$ function in isolation in *Toxoplasma*, but instead act in concert with each other to maintain appropriate basal levels of $Na^+$ in the cytosol. Reducing $Na^+$ intake through TgPiT deletion would impact the export activities of the $Na^+$ homeostatic pathway, such as reducing the ATP4-mediated extrusion of cytosolic $Na^+$. A reduced ATP4 activity would subsequently diminish the proton load in the cytosol, creating an alkaline environment. In parallel, ΔTgPiT parasites may also increase the activity of TgNHE1 to stimulate $Na^+$ uptake, leading to the expulsion of more $H^+$ from the parasite, and thus increased alkalinization of the cytosol. The loss of H+ may then be compensated by reducing the activity or expression of V-$H^+$-ATPase that exports $H^+$. To this point, the vacuolar ATPase subunit C (TGGT1_315620) is down-regulated in ΔTgPiT parasites.

ΔTgPiT parasites have a reduced cell volume. In protozoan species, PolyP, $P_i$ and sodium present in acidocalcisomes or in lysosome-like organelles (e.g., VAC, contractile vacuole, acidocalcisome-like vacuole) are key players in the osmoregulatory process [71–73]. These molecules are commonly used to restore the ionic strength of the cytosolic environment under hyperosmotic conditions, allowing the parasites to recover volume loss due to the movement of water down its concentration gradient. For example, polyP in *T. cruzi* are degraded or synthesized to recover cytosolic volume during hypoosmotic and hyperosmotic conditions, respectively [74]. Such an osmoregulatory process has never been fully demonstrated in *T. gondii*, however, the parasite harbors genes coding for a VTC complex involved in the synthesis of polyP [26] and an exopolyphosphatase responsible for the breakdown of polyP into phosphate monomers [16]. We show that TgPiT is an important $Na^+$-$P_i$ cotransporter in *Toxoplasma* both in cultured cells and in mice. At the plasma membrane, TgPiT imports $P_i$ from the parasite environment into the cytosol or the lumen of vesicles budding from the plasma membrane. The nature of these vesicles remains to be identified but they may correspond to acidocalcisomes that are rich in phosphate species. In *Toxoplasma*, it has been reported that acidocalcisomes fuse with the VAC to deliver their internal ionic and polyP content [40]. This fusion would likely allow the parasite's exopolyphosphatase activity to act upon the acquired polyP within VAC, breaking polyP down to monomeric $P_i$ and liberating any cations (e.g., $Na^+$, $Ca^{2+}$) forming complexes with polyP. By analogy to a phosphate transporter in *T. cruzi* localized to the contractile vacuole and expelling $P_i$ from this organelle into the cytosol, TgPiT

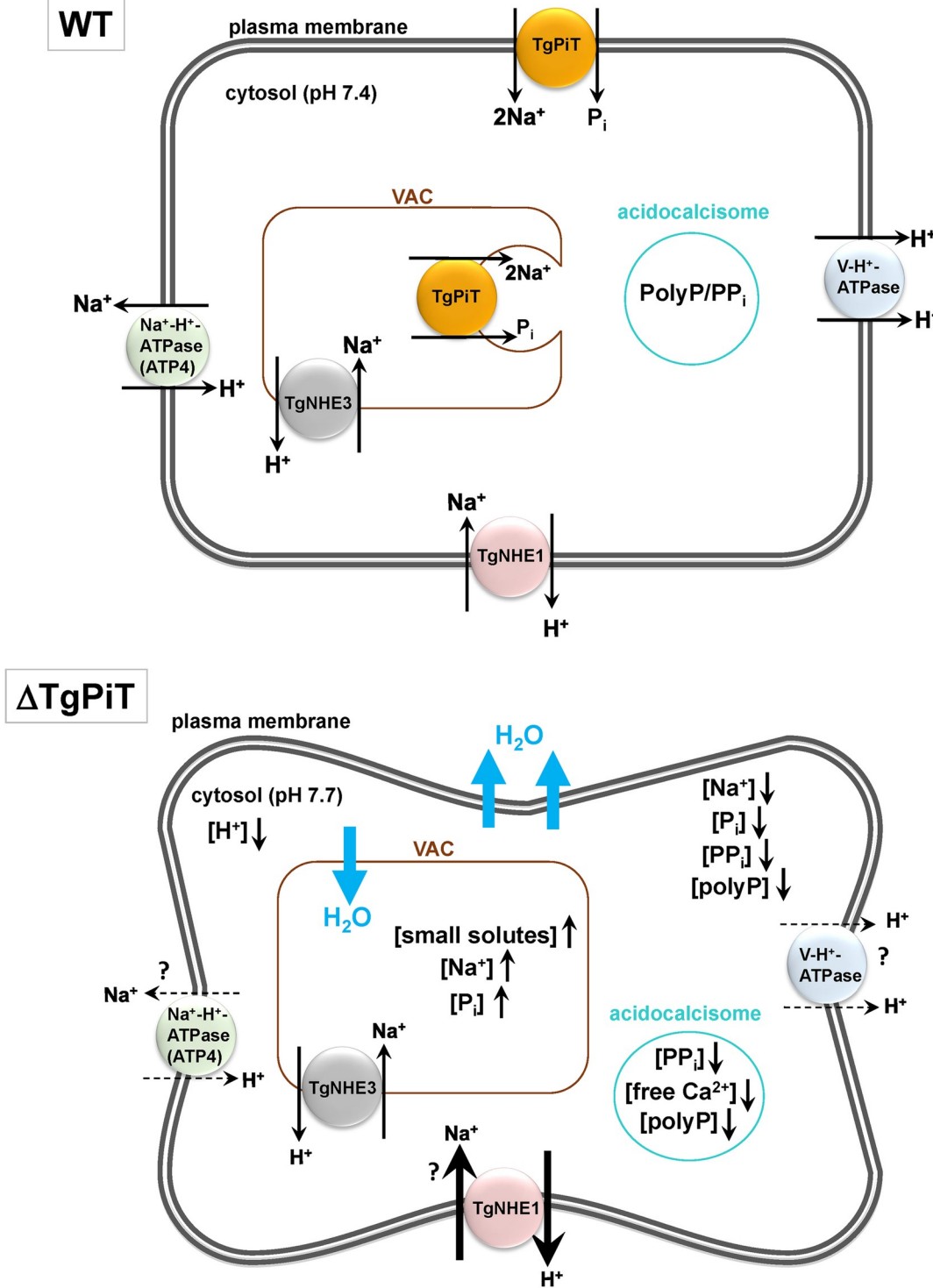

**Fig 14. Hypothetical model for ion homeostasis and osmoregulation in *Toxoplasma*.** Na$^+$ homeostasis in *Toxoplasma* is accomplished via cooperation between ATP4, TgNHE1 and TgPiT. When intake of Na$^+$ is reduced through TgPiT ablation, compensatory mechanisms may include decreased activity of ATP4 (dashed arrows) and increased activity of TgNHE1 (thick arrows), both resulting in reduced H$^+$ concentrations in the parasite and thus cytosolic alkalinization. Consequently, the mutant decreases the export of H$^+$ mediated by V-H$^+$-ATPase (dashed lines). ΔTgPiT parasites have a severe constitutive loss in cytosolic ionic strength through reduced internalization of P$_i$ and sodium into the parasite and impaired expulsion of these ions from VAC into the cytosol. Due to low P$_i$ availability, the mutant has reduced concentrations of total P$_i$ and polyP that is compensated for increased numbers of acidocalcisomes in an attempt to garner as much phosphate as possible.

in VAC may similarly function in exporting $P_i$ to the cytosol in *Toxoplasma*. The transfer of free $P_i$ and cations back into the cytosol could maintain overall ionic homeostasis during the progression of the parasite's lifecycle through drastically different intracellular and extracellular environments. Due to the dual localization of TgPiT at the plasma membrane and VAC, ΔTgPiT parasites may suffer from a severe constitutive loss in cytosolic ionic strength through a two-fold effect: reduced internalization of $P_i$ and sodium into the parasite's cytosol and impaired expulsion of these ions from VAC into the cytosol (Fig 14). Cytosolic reduction of $P_i$ and sodium in ΔTgPiT parasites leading to a loss in body volume, gives these mutants a phenotypically skinny appearance, likely due to the movement of water down its osmotic gradient away from the cytosol. Thus, in conjunction with a contribution to ionic homeostasis, phosphate is essential for cellular processes and communication, and the mechanisms of mobilization of $P_i$ are vital for providing *Toxoplasma* an adequate supply of phosphate for signaling, via phosphorylation, and ATP synthesis.

ΔTgPiT parasites exhibit defects in $Ca^{2+}$ release from acidic compartments, but the basal cytosolic $Ca^{2+}$ concentration remains unaltered. The maintenance of normal resting cytosolic $Ca^{2+}$ levels is crucial for *Toxoplasma* as the parasite relies on strict temporal fluxes of cytosolic calcium, and on sensitive triggers of such fluxes, to successfully progress through all stages of its lytic cycle. The importance of $Ca^{2+}$ homeostasis in *T. gondii* is further substantiated by the up-regulation of several calcium-mobilizing proteins in ΔTgPiT parasites based on our RNA-seq data. The mutant has reduced sensitivity to $NH_4Cl$-mediated $Ca^{2+}$ release, suggesting dysfunctions in the storage and/or release of $Ca^{2+}$ from acidic stores. $NH_4Cl$ typically causes an increase in cytosolic $Ca^{2+}$ through the alkalinization of acidic compartments that have $Ca^{2+}$-$H^+$ exchangers located on their membrane [25], thus inducing an exchange of luminal $Ca^{2+}$ for cytosolic $H^+$. A major acidic store of $Ca^{2+}$ in *T. gondii* is the acidocalcisome [49]. As the concentrations of free phosphate and calcium are inextricably linked, a reduction in $Ca^{2+}$ stored in acidocalcisomes in ΔTgPiT parasites may be attributed to low concentrations of phosphate species, specifically polyP that chelate and sequester $Ca^{2+}$ [46–48]. As polyP play an important role in osmoregulation [16], the control of cell shrinkage of the mutant would likely rely on the maintenance of higher amounts of polyP over monomeric phosphate to promote volume recovery, and thus favoring the sequestration of $Ca^{2+}$ on polyP instead of its liberation, further reducing $NH_4Cl$-mediated release of $Ca^{2+}$ from acidic stores (Fig 14). The significant reduction in calcium released from acidic organelles indicates that a large portion of calcium storage within these compartments is due to the presence of polyP.

ΔTgPiT parasites exhibit a swollen VAC. VAC is involved in protein degradation [75], and TgCRT located on this organelle functions as a small solute transporter [76, 77]. Parasites lacking TgCRT have an abnormally enlarged VAC due to the overaccumulation of molecules in VAC, leading to water influx [77]. It has been suggested that TgCRT plays a central role in controlling VAC size and integrity through the movement of small molecules (osmolytes) through the VAC membrane [77]. However, unlike ΔTgPiT parasites, ΔTgCRT mutants have a normal body size and shape (Z. Dou, personal communication), indicating TgCRT is not implicated in the maintenance of cytosolic ionic strength and overall osmoregulation of *Toxoplasma*, but solely in the movement of osmolytes from VAC into the cytosol to maintain VAC osmolarity. It remains conceivable that, to survive under constitutive hyperosmotic conditions, ΔTgPiT parasites crucially rely on activities of TgCRT and plasma membrane-localized $Na^+/H^+$ exchangers to sustain the ionic strength of the cytosol.

ΔTgPiT parasites have a reduction in overall polyP and $P_i$ content, yet concurrently expand the population of their acidocalcisomes, the main reservoirs of phosphate and polyP molecules. One possible explanation of this rather contradictory phenotype is that the increase in acidocalcisome number in response to diminished global phosphate availability may reflect a

form of acidocalcisome homeostasis and a feedback mechanism connecting phosphate content and sensing to acidocalcisome biogenesis. The supranumerous acidocalcisomes formed by ΔTgPiT parasites can then be used for two purposes: first, to efficiently garner enough phosphate (e.g., liberated from any phosphometabolites) to meet the basal demands of the parasite and second, to mobilize polyP as buffers to protect the parasite against stress osmotic conditions. Such a protective role of polyP has been previously reported for yeast and protozoan species when extracellular phosphate concentrations are limited or $P_i$ uptake compromised [45, 73, 78]. Regardless of the exact activities for phosphate acquisition and storage occurring in the acidocalcisomes of the mutant, these parasites suffer from overall reduced $P_i$ and polyP availability, with limited capacity of each individual acidocalcisome to produce enough phosphate molecules to satisfactorily perform its osmoregulatory function of restoring the cytosolic ionic strength.

In addition to their abundance in ΔTgPiT parasites, acidocalcisomes display a great diversity in shape, size and e-density. This miscellany may represent snapshots of acidocalcisomal flux occurring in response to volumetric changes and ionic deficiencies of the mutant. It may also reflect different states in acidocalcisome biogenesis and their metabolic activities, such as the synthesis, breakdown or accumulation of polyP and other molecules. Larger acidocalcisomes may contain higher amounts of phosphate and/or broken down metabolites that accumulate in the lumen. Regardless, the loss of TgPiT and the concomitant stimulation in acidocalcisome biogenesis would subsequently lead to a heterogeneous population of acidocalcisomes, some of which with increased solute content and osmotic strength, similar to the VAC, causing water to flow from the cytosol into these organelles. Although it remains possible that TgPiT localizes to vesicles related to acidocalcisomes, its ascertained localization to VAC membranes reveal that internal stores of phosphate needed for the restoration of cytosolic ionic strength are ultimately destined for VAC organelles, and transport back into the cytosol via TgPiT. The dual localization of TgPiT at the plasma membrane and VAC organelles, and its vesicular transport in the cytoplasm reflect the important roles of this transporter related to various ion handling in *Toxoplasma*.

## Materials and methods

### Ethics statement

This study was performed in compliance with the Public Health Service Policy on Humane Care and Use of Laboratory Animals and Association for the Assessment and Accreditation of Laboratory Animal Care guidelines of Johns Hopkins University. The animal protocol was approved by Johns Hopkins University's Institutional Animal Care and Use Committee (MO10H42).

### Reagents and antibodies

All chemicals were obtained from Sigma (St Louis, MO) or Fisher (Waltham, MA) unless otherwise stated. Radioactive phosphorus-32 radionuclide ($^{32}P_i$, as orthophosphate, 1 mCi at Spec. Act.: 8500–9120 Ci/ml) was purchased from Perkin Elmer (Waltham, USA). Cipargamin ($C_{19}H_{14}Cl_2FN_3O$) was obtained from MedChemExpress LLC (Monmouth Junction, NJ). Fura-4F-acetoxymethylester (AM) ($C_{43}H_{44}FN_3O_{24}$) was purchased from Thermo Fisher Scientific (Waltham, MA) and bis-carboxyethyl-carboxyfluorescein (BCECF)-AM ($C_{42}H_{40}O_{21}$) from Cayman Chemical Company (Ann Arbor, MI). Secondary antibodies used for immunofluorescence conjugated to Alexa488 or Alexa594 were purchased from Invitrogen (Carlsbad, CA). The following primary antibodies were used: rabbit polyclonal anti-GRA7 antibody [79], mouse monoclonal anti-Hsp70, mouse polyclonal anti-αtubulin and mouse monoclonal anti-

SAG1 (gift from J.-F. Dubremetz, Université of Montpellier) and mouse polyclonal CPL (gift from V. Carruthers, Michigan University).

## Cell lines, parasites and culture conditions

Human foreskin fibroblasts (HFF) obtained from the American Type Culture Collection (Manassas, VA) were grown as monolayers and cultivated in α-minimum essential medium (αMEM) supplemented with 10% fetal bovine serum (FBS), 2 mM glutamine and penicillin/streptomycin (100 units/ml per 100 μg/ml), and maintained at 37°C in 5% $CO_2$ unless specified otherwise. For some assays, cells were incubated in the phosphate-free DMEM (Gibco, Gaithersburg, MD). The tachyzoites from the RH strain (Type I lineage) were used throughout this study and propagated *in vitro* by serial passage in monolayers of HFF as described [80]. For engineering of PiT-deficient parasites, the strain ΔK80ΔHXGPRT obtained from NIAID Reagent Program (Germantown, MD) was used.

## $P_i$ transport assays

5x10E6 extracellular *Toxoplasma* freshly egressed from HFF were incubated at 37°C for 10–30 min in a reaction mixture (0.2 ml) containing, unless otherwise indicated in the figure legends, 140 mM NaCl (or 140 mM choline chloride to study the $Na^+$-independent component of $P_i$ transport), plus 1.5 mM $CaCl_2$, 5 mM KCl, 10 mM HEPES-Tris (pH 7.4), 1 mM $MgCl_2$ and 100 μM $^{32}P_i$ (2.5 mCi/μmol). The pH was adjusted to 7.4 with a few microliters of HCl 37% before each assay. The addition of $^{32}P_i$ initiated the transporter assay, and after specific time points, the total volume of the parasite suspension was applied to nitrocellulose membranes of 0.22 μm (Millipore, Burlington, MA) into a 10-fold filter manifold system (Hoefer, Holliston, MA), following an extensive washing with an ice-cold solution containing 140 mM choline chloride, 1.5 mM $CaCl_2$, 5 mM KCl, 10 mM HEPES 10 mM Tris pH 7.4, 1 mM $MgCl_2$. Incorporation of $^{32}P_i$ into the cells was measured using a scintillation counter. To obtain the blank uptake values, parasites were exposed to a non-radioactive $P_i$ reaction mixture and then kept on ice for 60 min [81]. To determine the effect of pH on $P_i$ uptake, the same buffer described above was used, with the addition of 10 mM HEPES/MES/Tris at various pH values. The transport affinity ($K_{0.5}$) and maximal rate ($V_{max}$) were determined by measuring $^{32}P_i$ uptake at $P_i$ concentrations ranging from 5 to 500 μM. The equation of Michaelis-Menten was applied to determine the parameters of the $P_i$ concentration-dependence for $P_i$ uptake:

$$v\ P_i\ \text{uptake} = V_{max} \times [P_i]/\{(K_{0.5,Pi} + [P_i]\}$$

When the effect of $Na^+$ concentration on $P_i$ uptake was analyzed, the $P_i$ concentration was kept constant (100 μM) and the NaCl concentration was varied from 0 (nominal) to 50 mM. The analysis of $Na^+$ concentration-dependence on $P_i$ uptake was performed using the following the equation:

$$v\ P_i\ \text{uptake} = v_{chol} + \{V_{max} \times [Na^+]^{nH}/(K_{0.5,Na})^{nH} + [Na^+]^{nH}\}$$

where $v_{chol}$ is the velocity in the absence of $Na^+$ (140 mM choline chloride) and $n_H$ is the Hill coefficient.

Choline chloride was used to replace NaCl to maintain the medium osmolarity at ≈300 mOsM, and cipargamin (at 5, 50 and 500 nM) was used to collapse the $Na^+$ gradient. In order to assess the effect of perturbation of the parasite plasma membrane potential gradient on $P_i$ uptake, carbonylcyanide-*p*-trifluoromethoxyphenylhydrazone (FCCP; 1 μM) or bafilomycin A1 (10 nM) were used. For each experiment involving $^{32}P_i$ uptake, paraformaldehyde-treated

parasites were used as negative controls and cpm values associated with killed parasites (that corresponded to <5%) were subtracted from radioactive values associated with live parasites.

## Sequence analysis

Nucleotide and amino acid sequences were searched against the *T. gondii* database ([www.toxodb.org](www.toxodb.org)) and the NCBI database using the BLAST algorithm [82]. Multiple sequence alignment was created using ClustalW, and the resulting similarities were then visualized by subjecting the alignment to Boxshade ([www.ch.embnet.org](www.ch.embnet.org)). Percent identity and similarity were calculated using standard tools for sequence analysis from NCBI (ncbi.nlm.nih.gov).

## Cloning of full-length cDNA encoding TgPiT

The ORF of TgPiT was amplified from a *T. gondii* cDNA library (provided by Vern Carruthers, University of Michigan) using the primers FTgPiTUTR and RTgPiTUTR (see supplementary material S1 Table for the sequences of primers used in this study). Amplified fragments (2.6 kb) were subcloned into the pJET1.2/blunt vector using High Fidelity DNA Polymerase (Roche Applied Science, Indianapolis, IN), and the insertion was confirmed by enzymatic digestion and sequencing. To clone *PiT* in fusion with *mCherry* and generate the PiTmCherry vector, the pJET vector containing the PiT gene was reamplified from the pJET vector using the primers FTgPiTpTUB and RTgPiTpTUB, and the PCR product was then cloned into the BglII-AvrII site of the Linker mCherry vector (received from M-J. Gubbels, Boston College).

## Expression of recombinant peptide from TgPiT in *E. coli* and affinity purification

The coding sequence corresponding to peptide from amino acids 275 to 417 was PCR-amplified using the primers FPiTantiPep1 and RPiTantiPep1 and cloned into KpnI and AvrII sites of the pET-47b vector (Novagen, Madison, WI) to generate N-terminal 6-His tagged fusion protein of 17.8 kDa the The recombinant peptide expressed in *E. coli* M15 strain was purified under denatured condition on $Ni^{2+}$-NTA resin according to the Qiagen protocol. After purification, the peptides were refolded by diluting the sample 10 times in the refolding buffer (50 mM Tris/HCl pH 7.5, 1 mM EDTA, 1 M L-arginine, 1 mM reduced form of glutathione, 0.8 mM of oxidized form of glutathione) at 4°C overnight. The samples were then concentrated and dialyzed against phosphate-buffered saline (PBS). Another coding sequence corresponding to peptide from amino acids 277 to 310 was PCR-amplified using the primers FPiTanti-Pep2 and RPiTantiPep2 and cloned into the BamHI and HindIII sites of the pQE-30 vector (Qiagen, Hilden, Germany) and expressed similarly to generate N-terminal 6-His tagged fusion protein of 5.25 kDa. The recombinant peptide expressed in *E. coli* M15 from Qiagen.

## Generation of antibodies against TgPiT$_{275-417}$ and TgPiT$_{275-310}$

To monitor TgPiT expression, rat (Covance, Berkeley, CA) and mouse (in house) polyclonal antisera were generated using recombinant TgPiT$_{275-417}$ and TgPiT$_{275-310}$, respectively. Rat sera containing antibodies against TgPiT$_{275-417}$ were affinity-purified against the recombinant TgPiT peptide according to the protocol by AminoLink Kit (Pierce biotechnology, Rockford, IL, USA). For TgPiT$_{275-310}$ antibody production, mice were intraperitoneally injected twice, two weeks apart, with 10 μg of purified peptides in PBS mixed with AddaVax (InvivoGen, San Diego, CA) (1:1 ratio, v/v) in a 100 μl volume. Mouse immune sera were collected from the animals two weeks after the last peptide injection.

## Immunoblot analyses of parasites

For immunodetection of TgPiT, parasites were lysed by suspension in SDS gel-loading buffer (50 mM Tris-HCl (pH 6.8), 50 mM 2-mercaptoethanol, 2% SDS, 0.1% bromophenol blue, 10% glycerol) followed by boiling in a water bath. The samples were subjected to SDS-PAGE, and the proteins were then electrophoretically transferred to a membrane (Immobilon Transfer Membranes, Millipore, Bedford, MA). The membrane was immersed in blocking buffer (PBS containing 3% skim milk) for 60 min, and then incubated with rat anti-TgPiT (1:1000) or mouse anti-PiT antibodies (1:500) in the blocking buffer for 60 min. Unbound antibody was removed by washing the membrane six times with blocking buffer. Next, the membrane was incubated with horseradish peroxidase-conjugated goat anti-mouse IgG antibody (Amersham Pharmacia Biotech; dilution, 1:10,000) in blocking buffer for an additional hour, before detection by chemiluminescence using ECL-Plus.

## Analysis of the biochemical properties of TgPiT

The solubilization of TgPiT was analyzed using cell fractionation and TritonX-100 partition on extracellular *Toxoplasma*. Parasites were purified from fibroblast by 3-micron filtration and washed twice with ice-cold PBS by centrifugation at 480g for 10 min. The parasite pellet was placed on ice, resuspended in 0.5 ml of lysis buffer made of 150 mM NaCl, 1% TritonX-100 and 50 mM Tris-Cl (pH 7.4) prechilled to 4˚C per $5\times10E7$ parasites and incubated for 15 min on ice. After vortexing, the lysate was centrifuged at 20,000g for 10 min at 4˚C, and the supernatant and pellet were used for TgPiT and SAG1 detection by Western blotting. The surface-exposure of TgPiT was assessed by mild proteolysis of extracellular parasites using proteinase K. Extracellular parasites (10E9/ml) were PBS-washed and resuspended in reaction buffer made of 50 mM Tris–HCl (pH 8), 150 mM NaCl, 5 mM $CaCl_2$, with 90 μl of parasite suspension used for each reaction. 10 μl reaction buffer (untreated control) or 10 μl of proteinase K was added to suspensions at the final concentration of 100 μg/ml at 22˚C. After 30 min incubation, 1 μl of phenylmethylsulfonyl fluoride (PMSF) was added to a final concentration of 2 mM to inactivate proteinase K. The parasites were then collected by centrifugation and washed twice with reaction buffer supplemented with 5 mM PMSF to remove excess proteinase K. The supernatant and pellet were used for TgPiT and SAG1 detection by Western blotting.

## Real-Time qPCR for transcriptional analyses for TgPiT

Confluent HFF were infected with *Toxoplasma* for 30 min, followed by PBS washes to remove extracellular parasites then incubated in culture medium, with low or normal $P_i$ concentrations for 24 h or 48 h. For each time points and experimental conditions, RNA was extracted and cDNA synthesized using SuperScript III First-Strand Synthesis System and oligo(dT) primers. qPCR amplification of cDNA (100 ng each) was performed using 2.25 μl of each of the forward and reverse primer pairs (10 μM) and 12.5 μl of PowerUp SyBr Green Master Mix (Applied Biosystems, Foster City, CA). Reactions were run on the StepOne Plus Real-Time PCR Systems (Applied Biosystems). The data of TgPiT was normalized to parasite α-actin housekeeping gene and $2^{-\Delta\Delta CT}$ values were calculated. The different sets of primers for the PiT gene were used: FqPCRTgPiTP1 and RqPCRTgPiTP1, and FqPCRTgPiTP2 and RqPCRTgPiTP2. Primers for the α-actin gene were Fα-actin and Rα-actin.

## TgPiT disruption using the CRISPR/CAS9 system

The knockout of TgPiT was accomplished using the CRISPR/CAS9-based system in *Toxoplasma* as previously described [83]. To create a double stranded break in the sequence

GAATACGTTCAGCAGTTCGG of the first exon of *PiT* in the *Toxoplasma* genome, we used the plasmid pSAG1::CAS9-U6::sgUPRT as a vector backbone, and the protospacer for *PiT* exon 1 was altered by using NEB Q5 Kit utilizing primers Q5(f)129 and Q5(r)129. The final construct was the CRISPR/CAS9 component of the PiT gene knockout strategy. To create the drug resistance cassette, a region corresponding to 920-bp nucleotides upstream of the *PiT* genomic sequence was amplified with primers F5'PiTKpnI and R5'PiTXhoI, and cloned into the KpnI-XhoI sites of pMiniHXGPRT. A region corresponding to 996-bp nucleotides of the 3' UTR sequence of the PiT genomic sequence was also amplified using the primers F3'PiT-BamHI and R3'PiTXbaI, and cloned into the BamHI-XbaI sites of pMiniHXGPRT. The construct that contains the 5' PiT UTR sequence, the HXGPRT drug resistance cassette and the 3' PiT UTR sequence, was digested with KpnI and XbaI, liberating a linear fragment. The drug resistant fragment was then co-transfected by electroporation with the CRISPR/CAS9 construct into the ΔKu80ΔHXGPRT strain of *Toxoplasma*. Transgenic parasites were selected with 25μg/ml of mycophenolic acid and 50μg/ml of xanthine. Two days post-transfection, surviving parasites were cloned by limiting dilution on HFF monolayers grown in 96-well plates. Genotype information for clonal lines generated were tested for the loss of the PiT gene by PCR and is listed in (S1 Table).

## TgPiT complementation of ΔTgPiT parasites

The complementation of the PiT gene was accomplished by integrating a cDNA copy of TgPiT to the UPRT locus on chromosome XI of ΔTgPiT parasites using the CRISPR/CAS9 system and the plasmid pSAG1::CAS9-U6::sgUPRT that has the protospacer for cutting exon 2 of the UPRT gene. To replace the UPRT locus with a cDNA of TgPit, a targeting construct containing the 5' UTR of UPRT, the cDNA of TgPiT and the 3' UTR of UPRT in a Puc19 vector was synthesized by combining 4 fragments using the NEB Builder DNA assembly kit: Fragment 1 consisted of 1000 bp of the genomic 5' UTR of UPRT amplified by using the primer pair 5'UTRfwd and 5'UTRrev; Fragment 2 was the TgPiT cDNA amplified the PiTmCherry vector with primer pair PiTcDNAfwd and PiTcDNArev; Fragment 3 corresponded to 1000-bp of the 3' UTR of UPRT, synthesized with primer pair 3'UTR fwd and 3'UTRrev; and Fragment 4 was a 2654 bp Puc19 segment, which containing the restriction sites HindIII and EcoRI for liberation of a linear targeting construct, was generated using the primer pair Puc19fwd and Puc19rev. Following the seamless synthesis of these four fragments, the resulting plasmid was digested with HindIII and EcoRI releasing a 3.6 kb targeting cassette. This targeting fragment that contains the TgPiT cDNA and the UTR sequences, was co-transfected with pSAG1:: CAS9-U6::sgUPRT into ΔTgPiT parasites. Two days post-transfection, transgenic parasites were selected with 5 μM 5-fluorodeoxyuridine, and surviving parasites were cloned as described above.

## Transcriptome sequencing

Parental and ΔTgPiT parasites were grown in fibroblasts for 24 h. The parasites were syringe-released by repeated passage through a 20 G1 needle then by a 22 G1 ½ needle, collected by centrifugation and resuspended in PBS buffer. Total RNA for the two strains from 3 independent parasite preparations was extracted using a Qiagen RNeasy kit. The total RNA was shipped to Novogene Corporation (Sacramento, CA) to be processed and analyzed. First, total RNA was converted to sequencing read libraries using the NEB Next Ultra kit (Illumina). The libraries were subjected to paired-end sequencing using the Novaseq 6000 with a read length of 150 bp. Each sample was sequenced to a depth of at least 20 million reads. The sequencing

reads per sample were trimmed and mapped to the genome of *Toxoplasma* GT1 strain (release 39) for gene differential expression.

## *In vitro* assays for parasite growth and replication

To monitor parasite development, standard plaque assays were performed using HFF grown until confluence in 6-well plates, infected with 150 *Toxoplasma* WT, parental or ΔTgPiT parasites per well and incubated at 37˚C for 7 days in culture medium (unless otherwise indicated), with low or normal $P_i$ concentrations. The cells were fixed and stained as described previously [80]. The plates were scanned (ScanWizard 5, Microtek) to calculate the average area of the plaques using Volocity software and graphed in Excel (Microsoft). *T. gondii* replication was assayed either by uracil incorporation assays or parasite enumeration per PV. For uracil assays, HFF were grown until confluency in 24-well plates prior to infection with 1×10E5 parasites for 24 h at 37˚C in medium with low or normal $P_i$ concentrations. Cells were then incubated with 1 μCi of [$^3$H]uracil for 2 h at 37˚C and the samples were processed as described [80]. To assess parasite number per PV, coverslips with confluent HFF were infected with parental or ΔTgPiT parasites for 4 h, thoroughly washed with PBS to remove extracellular parasites, and incubated in medium with low or normal $P_i$ concentrations. Coverslips were fixed with 4% formaldehyde plus 0.02% glutaraldehyde in PBS for 15 min and viewed with a Nikon E800 microscope. The number of parasites was recorded for at least 150–260 PV on the coverslip for each condition. The means of means for each condition is graphed as a percentage of all PV recorded with standard deviations using Excel (Microsoft).

## Mice and in vivo virulence assays

Six- to eight-week-old, outbred Swiss Webster mice purchased from The Jackson Laboratory (Bar Harbor, ME) were infected with 150 (intravenous route) or 50 (subcutaneous route) parental, ΔTgPiT or ΔTgPiT::*PiT* parasites diluted in PBS, or PBS. Five or six mice were used for each experimental group. The infected mice were monitored for symptoms daily.

## Measurement of total inorganic phosphate content

Freshly egressed *Toxoplasma* were collected from fibroblasts, filtered through a 3-micron nucleopore filter, centrifuged and washed 2× in BAG buffer (116 mM NaCl, 5.4 mM KCl, 0.8 mM $MgSO_4.7H_2O$, 5.5 mM glucose, 50 mM HEPES at pH 7.3), Parasites were lysed by sonication and their phosphate content extracted on ice with 1 M $KClO_4$ and vortexing as described [84]. The resulting lysate was divided into two identical parts: one part was incubated with recombinant PPX (exopolyphosphatase) at 37˚C for 1 h to degrade all phosphate species from short and long chain polyP to the measurable monomeric Pi; this sample also includes free monomeric $P_i$, and the other part was incubated with heat inactivated PPX and used to solely measure free monomeric $P_i$ content as described [21]. The concentration of liberated and free $P_i$ were colorimetrically quantified using 0.045% malachite green and 4.2% ammonium molybdate in 4 M HCl solution, which in the presence of $P_i$, creates a complex detected at 600 nm absorbance.

## Measurement of cell volume

Freshly egressed parasites were collected and filtered twice through a 3 μm Whatman nucleopore syringe filter. Parasites were then pelleted, washed and re-suspended in Ringer's buffer (155 mM NaCl, 3 mM KCl, 2 mM $Ca^{2+}$, 1 mM $MgCl_2$, 3 mM $NaH_2PO_4$-$H_2O$, 10 mM HEPES, 10 mM glucose) for counting based on volume gating. Counting was carried out on a Beckman

Z1 Coulter counter that was calibrated before each replicate using 10 μm$^3$ latex beads. Extracellular parasites were counted after gating on the basis of volumes ranging from 10 μm$^3$ to 40 μm$^3$. Three independent experiments were carried out for each parasite strain yielding 3 standard curves in S6 Fig. Each standard curve represents a decline in parasite number as the gating volume increases. For each biological replicate, the total number of parasites counted at the 10 μm$^3$ gate (our lowest gate) was considered 100% of parasites (our denominator). All subsequent counts from larger gates (our numerators) were normalized as percent of population decline. All parasites were determined to have a volume exceeding 10 μm$^3$ and readings at each gating volume were performed on a single sample preparation for each experiment. After normalization, a linear regression analysis was performed on every curve and the resulting equations were averaged and used to determine the mean volume of the parasite.

## Measurement of intracellular pH

Intracellular pH was measured fluorometrically using the ratiometric fluorescent, pH sensitive dye BCECF- AM. Extracellular parasites were harvested, passed through a 3 μm filter, resuspended to a final concentration of 1×10E9 parasites/ml, and then loaded with 9 μM BCECF-AM in Ringers buffer containing 1.5% sucrose at 37˚C for 20 min. After loading, parasites were washed twice in Ringer's buffer and resuspended to the same concentration used for loading. Fluorescent measurements were conducted using a 50 μl aliquot of parasite suspension diluted in 2.45 ml of Ringer's buffer to a final concentration of 2×10E7 parasites/ml in a cuvette. The cuvette containing our parasite suspension was left to equilibrate for 2 min before being placed into a Horiba Fluorolog high performance spectrophotometer. Fluorescence measurements were carried out with excitation occurring at 505 and 440 nm, and emission at 530 nm. Measurements were taken every 0.6 sec. The measured ratio of BCECF-AM fluorescence was correlated to pH values by generating a standard curve for each loading event for each cell line (n = 3): parasites were exposed to 0.1% TritonX-100 to lyse parasites and expose intracellular BCECF-am to Ringer's buffer at pH values ranging from 6.0 to 8.0, in 0.5 pH increments. The pH of the buffer and the fluorescence ratio reached after stabilization of the tracing was used to correlate fluorescence ratio, specific to each loading event, to pH value. The standard curve was then used to determine the pH.

## Measurement of cytosolic calcium content

Parasites were collected and filtered through a 3 μm Whatman nucleopore syringe filter before centrifugation at 1200×*g* for 10 min. Parasites were washed, resuspended in Ringer's buffer without calcium salts to a concentration of 1×10E9 parasites/ml and exposed to 5 μM Fura4-F-AM at 26˚C for 26 min. Fluorescence measurements were monitored in a Horiba Fluorolog fluorescence spectrophotometer. Dual fluorescence excitation occurred at 340 and 380 nm, and emission was recorded at 510 nm every 0.6 sec. The ratio of 340/380 emission was recorded to create tracings. When indicated, 10 mM NH$_4$Cl were added to the parasite samples. The Fura-4 fluorescence response to cytosolic Ca$^{2+}$ ([Ca$^{2+}$]i) was calibrated from the ratio of 340/380 nm fluorescence values after detergent lysis of the parasites and liberation of intracellular Fura-4F-AM to known concentrations of calcium as described [85, 86]. Changes in cytosolic calcium concentration were calculated by comparing averaged values 25 sec before and after NH$_4$Cl addition [87].

## Fluorescence microscopy

Immunofluorescence assays (IFA) on infected HFF were performed as described previously [88] with a 5 min permeabilization step with 0.3% TritonX-100 in PBS following fixation with

4% formaldehyde (Polysciences, Warrington, PA) plus 0.02% glutaraldehyde in PBS for 15 min. Coverslips were mounted using ProLong Diamond Antifade Mountant (Life Technologies) to minimize bleaching during microscopy. Slides were viewed using an oil immersion plan Apo 100x (NA1.4) objective, a Zeiss AxioImager M2 and a Hamamatsu Orca-R2 digital camera. Optical z-sections with 0.1μm spacing were acquired with Volocity software. The xyz registry was corrected and the images were processed using an iterative restoration algorithm with a confidence limit of 100% and an iteration limit of 35. Images were cropped to individual PV and the level of co-localization of TgPiT and other markers was measured. The thresholds for co-localization were set by measuring the background fluorescence of each channel using a region of interest (ROI) with the mean intensity value set as the lower threshold. The Pearson's Correlation Coefficient (PCC) and positive and negative PDM (product of the difference from the mean) channels were calculated. The Pearson's correlation varies between -1 and 1 with a correlation of 0 meaning that there is no linear relationship between the two signals. Correlations of -1 or 1 indicate a perfect negative or positive linear relationship, respectively. The PCC values for TgPiT and GRA7 are close to 0 indicating no relationship between the two signals, i.e. no co-localization. The PDM channels show the product of the difference from the mean for each voxel (3D-pixel) for the 2 channels analyzed. Yellow voxels (PDM +) are where the signal from both channels (red and green) are above the mean and show a positive correlation while the purple voxels (PDM -) are where the signal from one channel is above the mean while the signal from the other channel is below the mean indicating a negative correlation.

## Electron microscopy

For ultrastructural observations of *T. gondii*-infected cells by thin-section transmission electron microscopy (EM), infected cells were fixed in 2.5% glutaraldehyde in 0.1 mM sodium cacodylate (EMS) and processed as described [89]. Ultrathin sections of infected cells were stained before examination with a Philips CM120 EM (Eindhoven, the Netherlands) under 80 kV. For quantitative morphometric analysis of VAC, 44 to 59 representative electron micrographs from parental and ΔTgPiT parasites were randomly selected at low magnification to ensure the entire parasite fit into the field of view and could be analyzed as described [90] by using the standard formula for randomly orientated cell and structures. For negative staining to enumerate acidocalcisomes, extracellular parasites were washed and allowed to adsorb on carbon-coated formvar grids for 10 min at room temperature, blotted dry, and observed directly by electron microscopy [16]. For immunostaining of TgPiT, infected cells were fixed in 4% paraformaldehyde (Electron Microscopy Sciences, Hatfield, PA) in 0.25 M HEPES (ph7.4) for 1 h at room temperature, then in 8% paraformaldehyde in the same buffer overnight at 4°C. They were infiltrated, frozen and sectioned as previously described [91]. The sections were immunolabeled with rat anti-TgPiT antibodies (1:20 in PBS/1% fish skin gelatin), then with mouse anti-rat IgG antibodies, followed by 10 nm protein A-gold particles.

## Supporting information

**S1 Fig. Alignment of the N- and C-terminal sequences of ScPHOA, TgPiT and PfPiT (see sequence information in the text).**
(PDF)

**S2 Fig. TgPiT localization in *Toxoplasma*.** Fibroblasts were infected with *T. gondii* for 24 h before immunostaining using antibodies against TgPiT and GRA7 (dense granules and PV membrane). The Pearson's Correlation Coefficient (PCC) and positive and negative product of the difference from the mean (PDM) channels were calculated. The negative PDM values

(purple voxels) indicate no association of TgPiT with dense granules or the PV membrane.
(PDF)

**S3 Fig. Strategy for *PiT* deletion and verification (see description in Materials and Methods section and primers in S1 Table. A.** Schematic depiction for *PiT* ablation from the *Toxoplasma* genome. **B.** Confirmation of the double homologous recombination by screening PCR using the primers shown in red (in B), by staining with anti-TgPiT antibodies on immunoblots of parasite lysates (in C; arrow showing a band at ~96 kD) and fixed extracellular parasites (in D), comparing the parental and knockout strains.
(PDF)

**S4 Fig. Strategy for complementation of ΔTgPiT parasites and verification (see description in Materials and Methods section and primers in S1 Table). A.** Schematic depiction for *PiT* reinsertion in TgPiT parasites. **B.** Confirmation of the double homologous recombination by screening PCR using the primers shown in red. **C.** Transcriptional profiles of TgPiT in the parental and complemented (TgPiT::*PiT*) strains using total RNA subjected to first-strand cDNA synthesis using oligo(dT) primers. Left: Absolute abundance of *PiT* transcripts in the parental and TgPiT::*PiT* strains. Right: Transcript abundance Log$_2$ fold-change calculated by normalization with the housekeeping gene transcript, Tgα-actin. Data are means ± SD (n = 3 independent assays). NS (*p* = 0.2; Mann-Withney U test).
(PDF)

**S5 Fig. Phenotype of TgPiT::*PiT* parasites. A.** Western blot of parasite lysates from the parental, ΔTgPiT and ΔTgPiT::*PiT* strains probed with anti-TgPiT antibody showing TgPiT expressed in complemented parasites. Loading control: *Toxoplasma* α-tubulin (Tgα-Tub; 51 kDa) probed with mouse anti-α-tubulin antibody. **B.** IFA showing expression of TgPiT in ΔTgPiT::*PiT* parasites using anti-TgPiT antibodies. **C-D.** Microscopic observation of PV size in parental and ΔTgPiT::*PiT* parasites at 24 h and 48 h p.i. and parasite enumeration per PV showing no difference between the 2 strains. Arrowheads point to egressing parasites. **E.** Plaque assays for 7 days of parental and ΔTgPiT::*PiT* parasites infecting fibroblasts and dotplots to quantify lysis area, showing growth restoration to normal levels for the complemented strain. Differences in replication (D) and growth (E) between the two strains were not statistically different (unpaired Student's *t*-test).
(PDF)

**S6 Fig. Generation of the standard curves used to determine the cell volume for parental, ΔTgPiT and ΔTgPiT::*PiT* parasites using a Coulter counter.** Panels a-c: Volume distribution of parasites for each biological replicate consisting of three technical replicates. Data are mean ± SEM. Extracellular parasites were counted after gating on the basis of 6 different volumes, from 10 μm$^3$ to 40 μm$^3$, in 5 μm$^3$ increments. The raw number of parasites counted at each gate was normalized to the raw number of parasites at the 10 μm$^3$ gate, representing 100% of the parasite population, for each strain. After normalization, a linear regression analysis was performed on each curve, and the resulting equations was averaged and used to determine the mean volume for each parasite line.
(PDF)

**S1 Table. List of primers used in this study.**
(PDF)

**S2 Table. List of genes that are up-regulated in ΔTgPiT parasites and identified in SwissProt Bank.**
(PDF)

**S3 Table. List of genes that are down-regulated in ΔTgPiT parasites and identified in SwissProt Bank.**
(PDF)

**S4 Table. List of genes that are up-regulated in ΔTgPiT parasites and identified in ToxoDB.**
(PDF)

**S5 Table. List of genes that are down-regulated in ΔTgPiT parasites and identified in ToxoDB.**
(PDF)

## Acknowledgments

The authors gratefully thank the members of the Coppens laboratory for helpful discussions during the course of this work, and the technical staff from the Johns Hopkins Microscopy Facility (M. Delannoy, B. Smith) and the Yale Microscopy Facility (K. Zichichi). We acknowledge the generous providers of plasmids and antibodies. We are also grateful to Viviana Pszenny (NIAID) for her help with the CRISPR/Cas9 constructs and techniques.

## Author Contributions

**Conceptualization:** Beejan Asady, Claudia F. Dick, Isabelle Coppens.

**Data curation:** Beejan Asady, Claudia F. Dick, Isabelle Coppens.

**Formal analysis:** Beejan Asady, Claudia F. Dick, Karen Ehrenman, Julia D. Romano, Isabelle Coppens.

**Funding acquisition:** Claudia F. Dick, Isabelle Coppens.

**Investigation:** Beejan Asady, Claudia F. Dick, Karen Ehrenman, Julia D. Romano, Isabelle Coppens.

**Methodology:** Beejan Asady, Claudia F. Dick, Karen Ehrenman, Tejram Sahu, Julia D. Romano, Isabelle Coppens.

**Project administration:** Isabelle Coppens.

**Resources:** Isabelle Coppens.

**Supervision:** Beejan Asady, Claudia F. Dick, Karen Ehrenman, Julia D. Romano, Isabelle Coppens.

**Validation:** Beejan Asady, Claudia F. Dick, Karen Ehrenman, Tejram Sahu, Julia D. Romano, Isabelle Coppens.

**Visualization:** Beejan Asady, Julia D. Romano, Isabelle Coppens.

**Writing – original draft:** Isabelle Coppens.

**Writing – review & editing:** Beejan Asady, Claudia F. Dick, Karen Ehrenman, Tejram Sahu, Julia D. Romano, Isabelle Coppens.

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
