## [Decision Letter · Decision Letter 0]

13 Apr 2020

Dear Dr Coppens,

Thank you very much for submitting your manuscript "A unique Na+-Pi cotransporter in Toxoplasma plays key roles in phosphate import and osmoregulation" for consideration at PLOS Pathogens. As with all papers reviewed by the journal, your manuscript was reviewed by members of the editorial board and by several independent reviewers. In light of the reviews (below this email), we would like to invite the resubmission of a significantly-revised version that takes into account the reviewers' comments.

Overall the reviewers are supportive of the work but they have raised a number of concerns. Importantly some of your claims are not supported by the data presented. You will have to address all the shortcomings and notably by including a dose dependent curve of oubain inhibition.

We cannot make any decision about publication until we have seen the revised manuscript and your response to the reviewers' comments. Your revised manuscript is also likely to be sent to reviewers for further evaluation.

Sincerely,

Dominique Soldati-Favre

Associate Editor

PLOS Pathogens

Xin-zhuan Su

Section Editor

PLOS Pathogens

Kasturi Haldar

Editor-in-Chief

PLOS Pathogens

orcid.org/0000-0001-5065-158X

Michael Malim

Editor-in-Chief

PLOS Pathogens

orcid.org/0000-0002-7699-2064

Reviewer's Responses to Questions

**Part I - Summary**

Reviewer #1: The manuscript by Dick et al. reports experiments on phosphate transport in T. gondii and the identification of a phosphate/sodium symporter gene (TgPiT), ortholog to PHO89 in yeast, which has a plasma membrane and intracellular localization. Ablation of TgPiT led to a reduced phosphate uptake and polyphosphate levels, changes in their morphology, enlarging of the lysosome-like organelle (VAC/PLV), enlarging and increase in the number of acidocalcisomes, decreased infectivity in vitro and in vivo, and upregulation and downregulation of the expression of numerous genes. The work appears as carefully done and is clearly written. I suggest minor revisions.

Reviewer #2: This is a comprehensive, worthwhile, and well conducted study that will be of great interest to those interested in parasite biology and membrane transport proteins. It provides significant new insights into the uptake, roles and requirements for phosphate in Toxoplasma. This reviewer supports its publication in PLoS Pathogens however there are some major errors of interpretation that need to be corrected first, as well as various issues that require more detail or clarification. Furthermore, some claims are not supported by the data and should be toned down.

Reviewer #3: The manuscript by Dick et al. describes characterization of Na+-Pi co-transporter in Toxoplasma gondii. Overall, the study reveals importance of Pi import into the parasite and its dependence on Na+ gradient to achieve this. An orthologue of this transporter in Plasmodium falciparum was described in 2006, so its existence in Toxoplasma is not necessarily unique as the title of the manuscript states. The manuscript has considerable strength with regards to cell biological aspect of the transporter’s localization and consequences of the ablation of the gene encoding it. A major shortcoming of the manuscript relates to its claim that the Na+ gradient required for Pi transport is maintained by a Na+-K+-ATPase. Specifically, the authors need to address the following major points:

**Part II – Major Issues: Key Experiments Required for Acceptance**

Reviewer #1: (No Response)

Reviewer #2: 1. I disagree strongly with the authors’ interpretation that “The Na+ gradient required for Na+:Pi cotransport is maintained by a Na+-K+-ATPase activity”. The effect of a single (very high) concentration of a single compound (ouabain) is not enough evidence to make such a statement. I would make three points: i. The Toxoplasma genome does not even encode a Na+/K+ ATPase. ii. There is genetic and physiological evidence in reference 67 that TgATP4 (which is not a Na+/K+ ATPase) is responsible for maintaining the Na+ gradient across the plasma membrane of Toxoplasma parasites. iii. The concentration of ouabain that caused partial inhibition of Pi uptake in this study is orders of magnitude higher than the concentrations of ouabain reported to inhibit Na+/K+ ATPases.

Unless the authors have evidence that Toxoplasma parasites have an omeprazole-sensitive H+-K+ ATPase on their plasma membrane, they should also tone down their assertions in relation to this pump.

Fig. 4B should be removed as there is insufficient evidence to propose such a model.

2. Looking at the images in Figs 10 and 11, it is not at all obvious to this reviewer that the cell body is on average skinnier, and the acidocalcisomes larger, in TgPiT knockout parasites compared to their parents. Can more data/analysis be provided if this claim is to be made? In this reviewer’s opinion there is not enough evidence in the study to “ascertain that TgPiT is essential for parasite…ionic homeostasis and osmoregulation” (P18) or to claim that “DTgPiT parasites have a reduced cell volume” (P20). More direct studies (e.g. measuring the concentrations of major cations, performing volume measurements with a Coulter Counter etc.) would be required to justify such strong statements.

Reviewer #3: 1. The requirement for an inward Na+ gradient for Pi import is clear and keeping with other members of PiT family of transporters. What is unclear is that this gradient is maintained by a Na+-K+-ATPase as claimed by the authors. Their claim is based on oubain inhibition of the pump, based on the use of a single dose of the inhibitor at concentration (2 millimolar) that extraordinarily high (oubain usually is used at nanomolar concentrations for mammalian cells). One could claim that the Toxoplasma pump may be less sensitive to the inhibitor, but there is no evidence for such a pump encoded by the genome. A dose dependent curve is essential for such an experiment, and must be included.

2. On the other hand, there already is a known Na+ pump encoded by all apicomplexans, originally identified in Plasmodium falciparum as PfATP4, which exchanges Na+ for a H+. Indeed, Lehane et al. have published work on TgATP4 (J. Biol. Chem. PMID: 30723156). The authors do not mention this pump in their manuscript as a likely generator of Na+ gradient. There are inhibitors of this pump being advanced as antimalarial drugs, but also have inhibitory effect on Toxoplasma as shown by Lehane et al. It would be important to have authors make use of one of these inhibitors to assess their effects on Pi transport. Again, they should provide a dose-dependent curve rather than using a single high concentration of the compound.

3. The claim for the existence of a H+-K+-ATPase is again based on the use of a single dose of omeprazole at a very high concentration. They are evoking this pump as a counter to the Na+-K+-ATPase, but again there is no evidence for such a pump being encoded in the parasite genome. If TgATP4 is the pump that builds Na+ gradient, there is no need for a K+ pump to be evoked for balancing ion concentrations.

4. In their localization studies, it is not clear whether all the images were from parasites expressing TgPiT from a transfected plasmid or some were from the endogenous locus. Inappropriate expression of a protein could lead to its mis-localization, so the authors need to be explicit for each of the images shown as to what parasites were used for imaging (both IEM and IFA).

**Part III – Minor Issues: Editorial and Data Presentation Modifications**

Reviewer #1: Major points

1. Figs. 1, 2, 3, and 4A could be combined in one panel

2. Indicate how the 2Na+:1H2PO4- stoichiometry was determined under M&M. There is little evidence on the presence of Na+-K+-ATPases or H+-K+-ATPases in unicellular eukaryotes and possible non-specific effects should be indicated. I suggest to move Fig. 4B to the supplemental information or delete it, as it is too speculative.

3. Were PCC calculated for Fig. 6B?

4. The changes in PCC values for Fig. 6G are small, and some statistical demonstration that the differences are real and not only observed in one cell is necessary.

5. Loading controls are missing for Figs. S3C, and S5A.

6. Was the PAM changed by mutagenesis when doing the complementation?

7. Use statistical test for the in vivo studies

8. Another explanation for the increase in size of acidocalcisomes is that polyphosphate hydrolysis releases the cations associated to it and these cations increase the acidocalcisomes osmolarity and water uptake. Biogenesis of more acidocalcisome would be compensatory.

9. Since the RNAseq data were not validated for any of the genes, the discussion of the upregulated and downregulated genes should be under Discussion.

Minor points

Page 4, last paragraph: PPi is a byproduct … PPi has also….

Page 6, first paragraph: absorbance values are given in mM concentrations. Explain method under M&M.

Page 4, second paragraph: use drive instead of energize. There is no energy expenditure.

Page 4, third paragraph: use sulfate instead of sulfur.

Page 7, third paragraph. A reference is needed for bafilomycin A1.

Page 9, second paragraph: use intracellular rather than intraparasitic. References 36,37 must be where VAC/PLV are first mentioned.

Page 9, third paragraph: move ref. 38 after diet. Define PCC the first time is mentioned.

Page 10, last paragraph: spell out UPRT

Page 15, third paragraph: ref 61 should be here. Define TbHMIT.

Page 17, last paragraph: a vacuolar ATPase subunit C (not synthase)

Figure 3B, b, Why is Pi different than the other two?

Figures S4C, and S5C-D: indicate meaning of error bars and n

Reviewer #2: 1. The first set of experiments in the results is not presented clearly or in enough detail.

i. “The absorbance values were 0.5 ± 0.03 mM”. 0.5 mM is not an absorbance value. Furthermore, could the authors please specify how many independent experiments were performed, what the error represents (SD? SEM?), and whether the data for parasites and fibroblasts were significantly different from each other (with statistical test specified).

ii. On first read I thought the authors were describing transport experiments (“Toxoplasma actively accumulates exogenous Pi against a concentration gradient”). Please rephrase this section to avoid confusion.

iii. In the Introduction it was stated that cells accumulate Pi to millimolar concentrations, and the authors’ results with fibroblasts are not consistent with this statement. Furthermore, the authors should interpret their results cautiously, as the Pi concentration inside ‘quiescent fibroblasts’ may not be the same as the Pi concentration in the cytosol of Toxoplasma-invaded fibroblasts.

iv. Please provide more detail on how these experiments were performed in the Methods section (time point, confirmation that standard curves were performed etc.).

2. The results section on the influence of pH on Pi uptake is not clear and needs revision. “These data suggest that the Pi transport mechanism of Toxoplasma has a higher affinity for H2PO4- import than for HPO42- at physiological pH”. Presumably, the most logical interpretation of the data is that the transporter has a strong preference for H2PO4(-) over HPO4(2-). The apparent variation of Km(Pi) with pH reflects the abundance of H2PO4-, not an actual variation with pH of the affinity by which anything binds to the transporter. This point does not come through clearly.

3. There are several issues with the gene transcription section:

i. There is not enough detail in Fig 14A or in the relevant part of the Results text or Fig. legend to make sense of what Fig 14A means. The level of gene expression is different for 413 genes in the knockout, and for 28 genes in the parent, relative to what?

ii. "0.9-fold increase/0.5-fold increase/0.7-fold increase" in expression levels (P15). Do the authors mean to refer to log2 fold changes?

iii. Sulfate permease is a transporter, and its upregulation should be covered in the paragraph on transporters (P15).

iv. On P17 the description of what the V-type H+ ATPase does should be changed from “transmembrane movement of acid substances” to “transmembrane movement of H+ ions”.

4. In the title and elsewhere, describing TgPiT as a ‘unique’ transporter could be misinterpreted as implying that there is something unusual about it. It is a member of a well-characterised Pi transporter family. Could ‘unique’ perhaps be replaced with ‘single’?

5. Statements such as “the TgPiT protein is required for infectivity”/”essential for parasite survival, virulence…” are not correct, as infection does occur/some parasites do survive. Perhaps these statements could be rephrased – e.g. “required for a normal level of…”

6. The Figure 1 legend should specify which data are being compared in the statistical tests.

7. Fig 2C – the units for the sulfate concentration should be provided (in the same way as they are for Pi in Fig 2B).

8. Fig 3A legend: What was the statistical test and which data were being compared?

9. P8 – “TgPiT was mainly recovered in the solubilized fraction, in accordance to the presence of potential TMD of the parasite transporter” – this is not clearly written, and more explanation is required for readers not familiar with this type of experiment.

10. Fig 7A – were any statistical tests performed? (There are no asterisks on this panel).

11. Fig 7C legend – what do the scale bars represent?

12. Fig S4C – more explanation is required in the Figure and/or legend. It is not clear what the two different bar graphs are showing.

13. P11 – “Growth monitored by plaque assays reveals…(Fig. S5D)”. I think the authors mean Fig. S5E here, and that their explanation for what is in Fig. S5D is missing from the Results text.

14. Table I – what statistical test was used?

15. The Fig 11A legend implies that there are images of the complemented parasites, but these do not appear in the Figure.

16. P13: “However, at low phosphate concentration…”. This is confusing, as there is clearly a difference at both phosphate concentrations. Why are statistics for the normal phosphate condition not shown in Fig 12A?

17. Bottom of P13: “The more dramatic growth impairment…” – Has it been verified with statistical tests that the DIFFERENCE in growth between the two phosphate conditions is greater for the knockout parasites than for the parental parasites?

18. P14: “The morphology of VAC compartments…suggesting compromised endoproteolytic activities…” – No data are provided for the morphological variability of the parents, so how is it possible for the reader to ascertain whether morphological variability is abnormal or not? Also, how does variable morphology suggest an increase in ionic strength? The logic is not clear. “Some acidocalcisomes were also seen closely apposed to VAC structures. This suggests that loss of TgPiT severely alters the morphology and content of the VAC…”. Again, the logic is not clear to this reviewer.

19. It is misleading to conclude that “TgPiT has a higher affinity for Pi than PfPiT” (P19), as it has been shown previously that the affinity of PfPiT for Pi increases with increasing [Na+], and the authors of this study only examined Pi affinity at a single [Na+] that was higher than any of those used when assessing the Pi Km of PfPiT.

20. P23. As written it appears that the solutions used for measuring Pi uptake at defined pH values did not actually have their pH values determined. The pH of the saline solution should be measured, not just the pH of the buffering agent being added.

21. P23. With regard to the radiolabelled Pi uptake measurements – it is possible that some radiolabelled Pi becomes bound to the outside of cells/trapped between cells during the experiments. It would have been good to estimate this extracellular radioactivity by doing measurements with a very rapid time point and in the presence of a high concentration of unlabelled phosphate (e.g. 500 uM) (to minimise the transport of radiolabelled Pi into the cells). Since this does not appear to have been done, it is difficult to know how much radiolabelled Pi was actually taken up INTO the cells in the absence of Na+. Perhaps a cautionary note about this issue could be added to the Methods section or the Fig. 1 legend.

22. P27. What does “the parasites were syringed” mean?

Reviewer #3: Depending upon the results obtained from the experiments suggested above, Figure 4B should be revised.

PLOS authors have the option to publish the peer review history of their article (what does this mean?). If published, this will include your full peer review and any attached files.

Reviewer #1: No

Reviewer #2: No

Reviewer #3: No
---

## [Decision Letter · Decision Letter 1]

13 Sep 2020

Dear Dr. Coppens,

Thank you very much for submitting your manuscript "A single Na+-Pi cotransporter in Toxoplasma plays key roles in phosphate import and control of parasite osmoregulation" for consideration at PLOS Pathogens. As with all papers reviewed by the journal, your manuscript was reviewed by members of the editorial board and by several independent reviewers. The reviewers appreciated the attention to an important topic. Based on the reviews, we are likely to accept this manuscript for publication, providing that you modify the manuscript according to the review recommendations.

Importantly reviewer #2 has remaining concerns.  Measuring intracellular Na+ concentrations is difficult, and looking at the data, the experiments do not appear to have worked well.  This reviewer considers that the data on intracellular Na+ concentrations are not absolutely essential for this paper, and that it would be better to remove them than to publish them in their current form.

Sincerely,

Dominique Soldati-Favre

Associate Editor

PLOS Pathogens

Xin-zhuan Su

Section Editor

PLOS Pathogens

Kasturi Haldar

Editor-in-Chief

PLOS Pathogens

orcid.org/0000-0001-5065-158X

Michael Malim

Editor-in-Chief

PLOS Pathogens

orcid.org/0000-0002-7699-2064

Reviewer Comments (if any, and for reference):

Reviewer's Responses to Questions

**Part I - Summary**

Reviewer #2: A number of improvements have been made to the manuscript, and the authors have performed further experiments to probe the physiological consequences of knocking out TgPiT. This reviewer does however have some concerns relating to some of the new data and how it is interpreted.

Reviewer #3: Authors have adequately addressed major points raised by this reviewer, especially showing that Na+ gradient necessary for Pi transport is susceptible to ATP4 inhibitor cipargamin. Overall, the manuscript provided solid evidence for the significance of TgPiT in biology of Toxoplasma.

**Part II – Major Issues: Key Experiments Required for Acceptance**

Reviewer #2: This reviewer’s main concern about the new data relate to the intracellular Na+ measurements. It is this reviewer's strong view that they should either be removed, or revisited.

It is not necessarily to be expected that removal of one Na+ entry pathway would reduce the resting Na+ concentration, as the level of ATP4 activity will vary with the [Na+].

In order to determine accurately whether or not there is a decrease in the already low resting Na+ concentration in the TgPiT knockout parasites relative to the parental parasites, it would be necessary to perform full calibrations in every experiment for every parasite line. Comparing Fluorescence Ratios for different parasite lines is simply not meaningful – these will be impacted by dye loading, parasite number etc. and can vary substantially between different experiments even for the same parasite line. A Fluorescence Ratio of 1 (the average for the TgPiT knockout parasites) is generally a sign that the experiment is not working (e.g. because the parasites have not taken up enough dye). The small, abrupt effect of cipargamin in these experiments is quite different to what has been observed in previous studies with T. gondii and P. falciparum, raising further concerns that these experiments are not working well.

Reviewer #3: (No Response)

**Part III – Minor Issues: Editorial and Data Presentation Modifications**

Reviewer #2: It is misleading to state, in the abstract or anywhere, that the TgPiT knockout parasites “are unable to regulate their cytosolic Na+ concentration and pH”. Clearly, the parasite’s primary Na+ regulator ATP4 will be functional, and will continue to regulate the parasite’s internal Na+ concentration such that it is kept much lower than the extracellular Na+ concentration. All parasites are also clearly regulating their pH, and maintaining intracellular pH values that are much higher than what would be the case if H+ ions were at electrochemical equilibrium across the parasite plasma membrane.

The pH experiments are not described in enough detail. What solution were the parasites suspended in when their resting pH was being determined?

The methods section for the Ca2+ measurements outlines a calibration procedure but the data are shown as Fluorescence Ratios. Can they be shown as [Ca2+]cyt instead? This would provide important information about whether the experiments were working well and is necessary to back up the statement that there is no difference in resting [Ca2+]cyt between the lines.

In future studies involving ion-sensitive fluorescent dyes, it would be preferable to calibrate using previously established methods involving the use of ionophores rather than by lysing parasites. The latter has the potential to affect the location, environment and behaviour of the fluorescent dyes.

The Coulter Counter experiment design seems unnecessarily complex. Would it be possible to include raw data from simple experiments measuring the volume distribution for the different parasite populations (graphs of number of cells on the y-axis and cell volume on the x-axis)? This would also provide information on whether there were differences among parasite lines with regard to the amount of variation in volume in the population.

Reviewer #3: (No Response)

PLOS authors have the option to publish the peer review history of their article (what does this mean?). If published, this will include your full peer review and any attached files.

Reviewer #2: No

Reviewer #3: No
---

## [Editor Report · Decision Letter 2]

14 Oct 2020

Dear Dr. Coppens,

We are very pleased to inform you that your manuscript 'A single Na+-Pi cotransporter in Toxoplasma plays key roles in phosphate import and control of parasite osmoregulation' has been provisionally accepted for publication in PLOS Pathogens.

Best regards,

Dominique Soldati-Favre

Associate Editor

PLOS Pathogens

Xin-zhuan Su

Section Editor

PLOS Pathogens

Kasturi Haldar

Editor-in-Chief

PLOS Pathogens

orcid.org/0000-0001-5065-158X

Michael Malim

Editor-in-Chief

PLOS Pathogens

orcid.org/0000-0002-7699-2064
---

## [Editor Report · Acceptance letter]

16 Dec 2020

Dear Dr. Coppens,

We are delighted to inform you that your manuscript, "A single Na+-Pi cotransporter in Toxoplasma plays key roles in phosphate import and control of parasite osmoregulation," has been formally accepted for publication in PLOS Pathogens.

Best regards,

Kasturi Haldar

Editor-in-Chief

PLOS Pathogens

orcid.org/0000-0001-5065-158X

Michael Malim

Editor-in-Chief

PLOS Pathogens

orcid.org/0000-0002-7699-2064